# Disrupting cortico-cerebellar communication impairs dexterity

**Jian-Zhong Guo**[1†]**, Britton A Sauerbrei**[1†]**, Jeremy D Cohen**[1‡]**, Matteo Mischiati**[1‡]**, Austin R Graves**[2]**, Ferruccio Pisanello**[3]**, Kristin M Branson**[1]**, Adam W Hantman**[1]*****

[1]Janelia Research Campus, Ashburn, United States; [2]Johns Hopkins University, Baltimore, United States; [3]Italian Institute of Technology, Genoa, Italy

**Abstract** To control reaching, the nervous system must generate large changes in muscle activation to drive the limb toward the target, and must also make smaller adjustments for precise and accurate behavior. Motor cortex controls the arm through projections to diverse targets across the central nervous system, but it has been challenging to identify the roles of cortical projections to specific targets. Here, we selectively disrupt cortico-cerebellar communication in the mouse by optogenetically stimulating the pontine nuclei in a cued reaching task. This perturbation did not typically block movement initiation, but degraded the precision, accuracy, duration, or success rate of the movement. Correspondingly, cerebellar and cortical activity during movement were largely preserved, but differences in hand velocity between control and stimulation conditions predicted from neural activity were correlated with observed velocity differences. These results suggest that while the total output of motor cortex drives reaching, the cortico-cerebellar loop makes small adjustments that contribute to the successful execution of this dexterous movement.

**\*For correspondence:**
hantmana@janelia.hhmi.org

[†]These authors contributed equally to this work
[‡]These authors also contributed equally to this work

**Competing interests:** The authors declare that no competing interests exist.

## Introduction

Motor cortex is a key brain region involved in the control of voluntary arm movement. In human patients, stroke and neurological disease affecting motor cortex lead to weakness or paralysis of the arm (*Krakauer and Thomas Carmichael, 2017*), and experimental perturbations of cortical function disrupt reaching in animal models (*Lawrence and Kuypers, 1968*; *Passingham et al., 1983*; *Fogassi et al., 2001*; *Miri et al., 2017*). Optogenetic silencing of excitatory neurons in the motor cortex of mice robustly blocks the initiation and execution of reaching (*Guo et al., 2015*; *Galiñanes et al., 2018*; *Sauerbrei et al., 2020*). Thus, motor cortical output is necessary for driving the arm toward the target. The optogenetic perturbations used in these studies, however, affect all channels of motor cortical output, including projections to other cortical areas, the spinal cord, the cerebellum, and the reticular formation. To test how cortical projections to specific targets influence downstream neural activity and behavior, it is necessary to selectively manipulate individual channels of cortical output. Projection-specific manipulations of cortical output have been challenging: optogenetic silencing of terminals can fail to block neurotransmission (*Mahn et al., 2016*), and optogenetic or electrical stimulation of cortical axons risks activation of collaterals to other targets. In the present study, we selectively disrupt cortical communication with the cerebellum, a major disynaptic target of cortical input.

Cerebral cortex communicates with the cerebellum (*Figure 1A–B*) through its monosynaptic projections to the pontine nuclei (PN), which include the basal pons and the reticulotegmental nucleus (*Cajal, 1898*; *Brodal and Jansen, 1946*; *Mihailoff et al., 1985*; *Brodal and Bjaalie, 1992*; *Leergaard et al., 2004*; *Legg et al., 1989*). The PN project to the cerebellum, their sole output target, providing mossy fiber inputs to the cerebellar cortex as well as collaterals to the deep cerebellar nuclei (DCN), the output stage of the cerebellum. The terminal fields of PN mossy fibers overlap with those of other mossy fiber systems; for instance, in the intermediate part of cerebellar cortex,

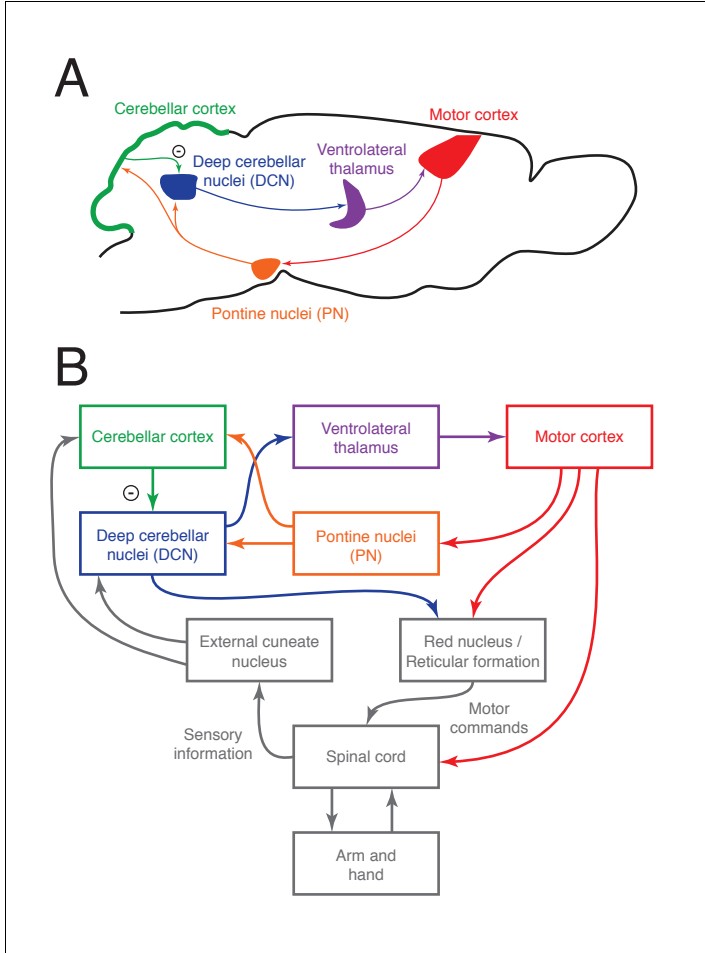

**Figure 1.** The cortico-cerebellar loop for arm and hand control in the mouse. (**A**) Schematic of the main anatomical connections within the loop. (**B**) Relationship between the cortico-cerebellar loop and the sensorimotor periphery.

they overlap with cuneocerebellar and spinocerebellar inputs carrying proprioceptive signals about the limbs. The major synaptic partners of mossy fibers are granule cells and Golgi cells. Granule cells excite Purkinje cells, which in turn inhibit the DCN. The DCN project to the ventrolateral thalamus, which projects back to motor cortex, closing the loop. The cerebellum also projects directly to lower motor centers in the pontomedullary reticular formation, red nucleus, and vestibular nuclei.

The cerebellum contributes to the coordination of arm movement. Cerebellar neurons, like motor cortical neurons, are strongly modulated during reaching (*Fortier et al., 1989*), though they exhibit more phasic firing patterns, weaker directional tuning, less holding-related activity, and higher trial-to-trial variability (*Fortier et al., 1993*). Cerebellar damage, disease, or perturbation results in timing deficits, ataxia, tremor, and dysmetria of voluntary arm movements, including reaching and grasping (*Dow and Moruzzi, 1958*; *Klockgether, 2000*; *Nashef et al., 2019*; *Becker and Person, 2019*; *Low et al., 2018*; *Mason et al., 1998*). These movement deficits are likely due to disruption of both descending cerebellar projections and ascending projections to cerebral cortex. In primates, disruption of cerebellar output by cooling or high-frequency microstimulation impairs arm movement and alters motor cortical activity (*Brooks et al., 1973*; *Meyer-Lohmann et al., 1975*; *Meyer-Lohmann et al., 1977*; *Nashef et al., 2019*). Arm movements are impaired even when outputs in the superior cerebellar peduncle are stimulated selectively (*Nashef et al., 2019*). During the planning of tongue movements in mice, cerebellar perturbation disrupts cortical activity, cortical perturbation disrupts cerebellar activity, and both disrupt motor planning (*Gao et al., 2018*). These previous studies have provided evidence that the cerebellum influences cortical activity related to

movement. The perturbations used in these studies, however, have been targeted to cerebellar regions which integrate input from both cortical and sensory pathways, and the effects of a specific disruption of cortical projections to the cerebellum are less clear.

Because the PN receive extensive cortical input and project only into the cerebellum, they provide an ideal target for testing how the cortico-cerebellar pathway influences neural dynamics and behavior. In primates, PN neurons are modulated during arm and eye movements and are tuned to movement direction (*Tziridis et al., 2009*; *Matsunami, 1987*). In humans, damage to sites overlapping with the PN results in motor impairments (*Schmahmann et al., 2004*), and the most prominent impairments from similar lesions in animal models have involved visual sensorimotor behaviors (*Stein and Glickstein, 1992*; *Levesque et al., 1986*) and gap crossing (*Jenkinson and Glickstein, 2000*). Optogenetic PN inhibition was recently shown to impact cortico-cerebellar coupling during arm movement (*Wagner et al., 2019*), but effects on behavior were limited to an increase in movement duration when the perturbation was applied in blocks of trials. Here, we use cell-type-specific optogenetic excitation to examine how PN output influences movement through its effects on downstream activity in the DCN and cortex. Our approach targets ChR2 expression to precerebellar PN neurons, which express VGLUT1, without interfering with the nearby corticospinal and medial lemniscus tracts, and without labeling nearby neurons outside the PN, which are VGLUT1-negative. We find that a selective disruption of cortico-cerebellar communication by optogenetic stimulation of the PN does not usually block the animal's ability to lift the limb and drive it toward the target, as a direct perturbation to motor cortex does. Instead, this perturbation affects the precision, accuracy, duration, and rate of success of the movement, suggesting a necessary role for the cortico-cerebellar loop in fine-tuning reach kinematics.

## Results

### Firing properties of PN neurons during a dexterous, cortically dependent behavior

To profile the activity of precerebellar PN neurons receiving input from motor cortex during reaching, we recorded from these cells during a cortex- and cerebellum-dependent (*Sauerbrei et al., 2020*; *Becker and Person, 2019*; *Guo et al., 2015*; *Galiñanes et al., 2018*) cued reach-to-grasp task (*Figure 2A*, upper left and lower). Because the regions of the PN that receive motor cortical input are small and deep in the brain, we targeted our recordings using a combination of high-density electrophysiology and optogenetic stimulation of corticopontine fibers (*Figure 2A*, upper right). Experiments were performed in Sim1-Cre X Ai32 mice, which express ChR2 in layer 5 pyramidal tract neurons projecting to the PN (*Gerfen et al., 2013*). During a recording session, we lowered a 960-site, 384-channel Neuropixels probe (*Jun et al., 2017*) coated in fluorescent dye toward the PN (*Figure 2B*). As the probe approached its target, forelimb motor cortex was stimulated with an optical fiber coupled to a 473 nm laser using a 1 s, 10 Hz pulse train. This resulted in bursts of spiking activity in the target zones, verifying that the probe was in a region of the PN receiving motor cortical input (*Figure 2C*).

We recorded the activity of 129 PN neurons, a subset of which (n = 30) exhibited responses to optogenetic stimulation of motor cortex within 20 ms of laser onset (right inset of neurons 2 and 3 in *Figure 2D*). To quantify when the earliest responses to the stimulation occurred, we performed a separate analysis based on spike counts in a sliding 10 ms window (see Materials and methods), and detected changes in spike counts in 5%, 25%, and 50% of responsive PN neurons at lags of 5, 7, and 9 ms, respectively (*Figure 2—figure supplement 1*). PN neurons exhibited a wide variety of patterns during the task (*Figure 2D–E*, *Figure 2—figure supplement 2*). In order to quantify the response of these cells to the cue and movement onset, we fit a generalized linear model (GLM) for each neuron (see Materials and methods). This produced two curves, one indicating how the cue influenced firing rate over time, and the other indicating the influence of movement onset. These curves are shown for three example neurons in *Figure 2F* (upper); these are the same neurons shown in *Figure 2D*. Neuron 1 exhibits a sharp, transient increase in firing rate following the cue, but little change around lift. Neuron 2 has an increase following the cue, but a decrease following lift. Neuron 3 shows a rapid and sustained increase following cue, and a smaller, delayed increase following lift. The cue and reach curves are shown for all neurons for which we were able to fit a GLM (defined as those cells

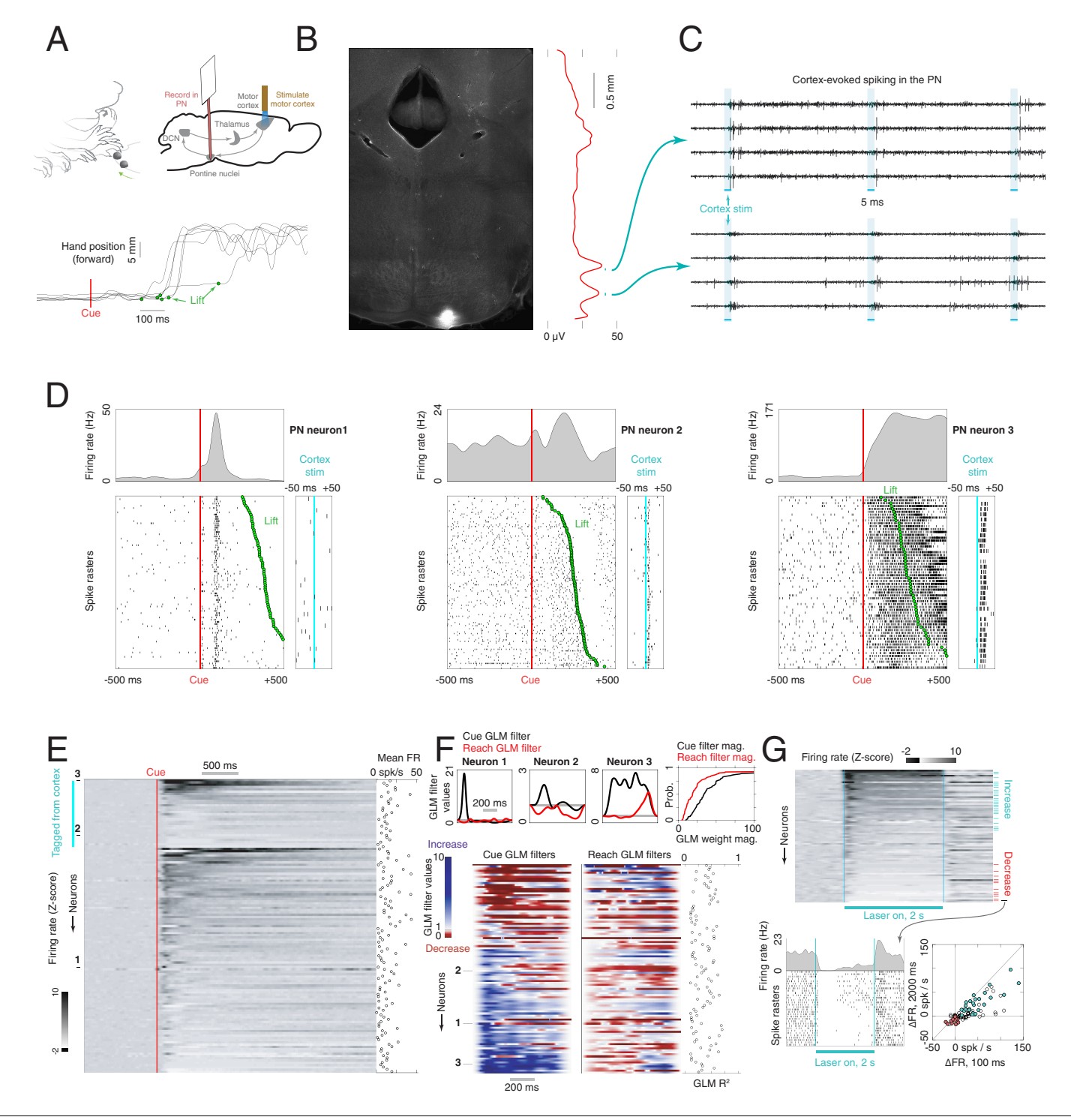

**Figure 2.** Firing properties of pontine nuclei (PN) neurons during a cortically dependent reach-to-grab task. (**A**) Experimental setup and reach kinematics. Upper left: mice were head-fixed and trained to reach to and grab a pellet of food following an acoustic cue. Upper right: strategy for recording from motor-cortex-recipient neurons in the pontine nuclei. An optical fiber was placed over the forelimb motor cortex of mice expressing ChR2 in pyramidal tract neurons (Sim1-Cre X Ai32). A 384-channel Neuropixels probe was lowered into the pontine nuclei as a laser pulse train was delivered to cortex. After these stimuli were delivered, the animal performed the task as PN neurons were recorded. Lower: example hand trajectories aligned to the cue. The green dots indicate lift, when the animal initiates the reach-to-grab sequence. (**B**) Histological section showing the tip of the probe in the PN. Red line shows the amplitude of multiunit activity on the probe in response to cortical stimulation as a function of depth. (**C**) Raw data from eight channels near the bottom of the probe showing activity evoked in the PN by motor cortex stimulation. (**D**) Spike rasters and firing rates for

*Figure 2 continued on next page*

*Figure 2 continued*

example neurons in the task, aligned to the cue. The timing of lift is indicated by the green dots. The panels on the right show each unit's response to cortical stimulation. (E) Firing rate z-scores for all PN neurons (n = 129), sorted according to whether the cells responded to cortical stimulation and by the magnitude of the response (n = 30 neurons were responsive). Inset at the right shows mean firing rates. (F) Effect of the cue and reaching on the firing rates of PN neurons (see Materials and methods). Upper: cue (black) and reach (red) components from generalized linear models (GLMs) for the three example neurons in (D). Gray horizontal lines at the value of 1 correspond to no effect on firing rate; values less than 1 correspond to a decrease, and values greater than 1 correspond to an increase. Lower: cue and reach components for all neurons with $R^2$ > 0.1 (n = 103). Blue values indicate firing rate increases, and red values indicate decreases. Inset at right shows $R^2$ for each neuron. (G) Response of PN neurons to a 2 s, 40 Hz stimulation of motor cortex. Upper: firing rate z-scores for each neuron, aligned to laser onset. Blue ticks indicate neurons with a sustained increase, and red ticks indicate neurons with a decrease. Lower left: laser-aligned spike rasters and firing rate for an example neuron suppressed by motor cortical stimulation. Lower right: scatterplot of the change in firing rate (FR) from baseline over the entire 2 s stimulation window vs. over the initial 100 ms of stimulation. Color coding as in the upper panel.

The online version of this article includes the following figure supplement(s) for figure 2:

**Figure supplement 1.** Timing of pontine nuclei (PN) responses to cortical stimulation.
**Figure supplement 2.** Firing rates of pontine nuclei (PN) neurons aligned to different behavioral events.

for which the $R^2$ from the regression of observed firing rates on the firing rates predicted by the GLM exceeded 0.1, n = 103/129) in *Figure 2F* (lower). Some cells were modulated around movement onset, consistent with previous results in primates (*Tziridis et al., 2009*; *Matsunami, 1987*), but the GLM weights for cue were typically larger than the weights for reach (*Figure 2F*, upper right; p = 2.2e−9, signed-rank test). These cue-aligned changes in firing rate might reflect acoustic responses, changes in attention, motor planning, or some combination of these factors. We did not observe a difference in the norm of the GLM weights between cortex-tagged and -untagged neurons for reach (p = 0.63, signed-rank test) or cue (p = 0.90, signed-rank test). Our observation that the activity of some motor cortex-recipient PN neurons is aligned both to the cue and movement suggests that these neurons might integrate signals of multiple modalities.

In rodents, there appear to be very few local inhibitory neurons within the PN (*Brodal et al., 1988*). Inhibitory inputs to the PN have been identified in the zona incerta, anterior pretectal nucleus, and reticular formation (*Border et al., 1986*), but it is unclear whether these circuits can be recruited by descending cortical signals to suppress PN spiking. In order to identify possible cortically driven feedforward inhibition onto PN neurons, we delivered a 2 s, sinusoidal laser stimulus to cortex at 40 Hz, a frequency near the upper limit for driving ChR2 (*Berndt et al., 2011*). Among the neurons tested with 2 s stimulation (n = 80), some maintained elevated firing rates during this long stimulation (n = 26/80), but others showed a sustained decrease (n = 12/80; *Figure 2G*, upper, lower left). In some cells, these decreases were not preceded by transient excitation; this suggests that the suppression does not reflect post-burst hyperpolarization but is instead likely due to feedforward inhibition (*Figure 2G*, lower right). Thus, motor cortex may be capable not only of exciting PN neurons through direct pyramidal tract projections, but also suppressing them through polysynaptic inhibitory routes.

## Propagation of PN signals across the cortico-cerebellar loop

How do the effects of PN stimulation propagate across the cortico-cerebellar loop? We addressed this question by optogenetically stimulating PN neurons and recording neural activity across the loop in awake animals at rest. Using the Slc17a7-Cre mouse line we generated (*Huang et al., 2013*) that expresses Cre recombinase selectively in VGLUT1 neurons, we drove expression of ChR2 in the PN by injecting AAV2/1-FLEX-rev-ChR2-tdTomato into these mice. A tapered optical fiber was implanted in the PN, exploiting its sub-micrometer implant cross-section and the smooth profile of the fiber, which make the targeted region easier to reach with respect to standard flat-cleaved fiber optics (*Pisanello et al., 2017*). We stimulated PN neurons with 473 nm light at 8–20 mW for 2 s at 40 Hz. We characterized the postsynaptic effects of stimulation by recording in the cerebellar hemispheres with a high-density Neuropixels probe (n = 2 mice). PN stimulation entrained multiunit activity in the cerebellar cortex at 40 Hz (*Figure 3—figure supplement 1*). This multiunit activity likely reflects combined signals from mossy fiber terminals, granule cells, Purkinje cells, and interneurons. Next, in a different set of animals, we recorded from individual Purkinje cells (n = 23 neurons, n = 2 mice) in the cerebellar hemispheres (*Figure 3A*, upper left). While many cells responded to PN

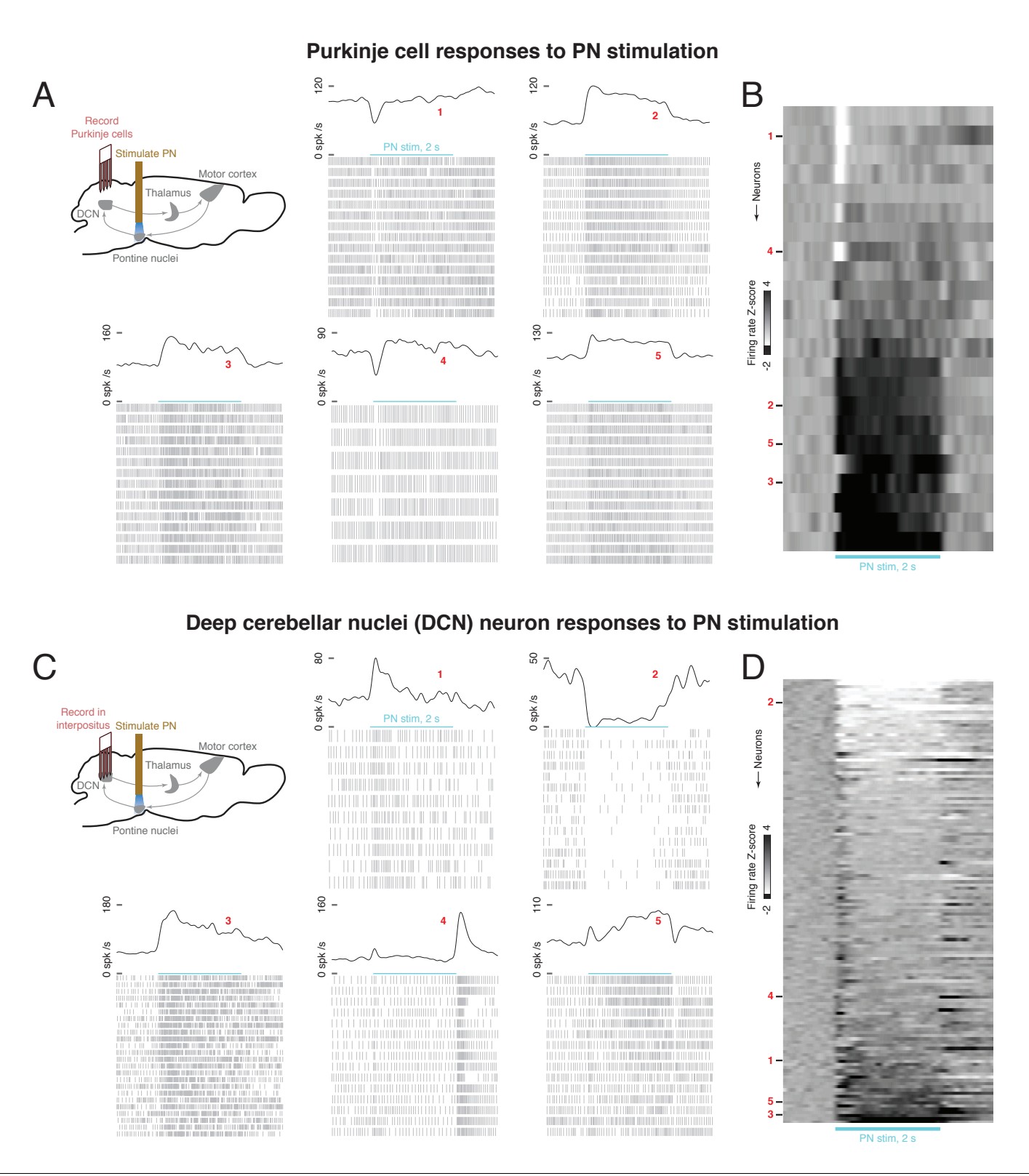

**Figure 3.** Effect of optogenetic stimulation of the pontine nuclei (PN) on Purkinje cell and deep cerebellar nuclei (DCN) activity. (**A**) Upper left: schematic of Purkinje cell recording experiment. A 2s, 40 Hz laser stimulus was applied to the PN while activity was recorded in cerebellar Purkinje cells. Lower and right: laser-aligned spike times and firing rates for five example Purkinje cells. Because the firing rates are high, every third spike is shown in the raster plots, for visual clarity. (**B**) Firing rate z-scores for all Purkinje cells (n = 23 neurons, n = 2 mice), aligned to PN stimulus onset. (**C**) Upper left:

*Figure 3 continued on next page*

*Figure 3 continued*

schematic of DCN recording experiment. A 2 s, 40 Hz laser stimulus was applied to the PN while activity was recorded in the DCN. Lower and right: laser-aligned spike times and firing rates for five example DCN neurons. For visual clarity, every third spike is shown in the spike rasters. (D) Firing rate z-scores for all DCN neurons (n = 139 neurons, n = 8 mice), aligned to PN stimulus onset.

The online version of this article includes the following figure supplement(s) for figure 3:

**Figure supplement 1.** Effect of optogenetic stimulation of the pontine nuclei (PN) on multiunit activity in the cerebellum.

**Figure supplement 2.** Entrainment of Purkinje cell and deep cerebellar nuclei (DCN) spikes to sinusoidal pontine nuclei (PN) stimulation.

**Figure supplement 3.** Histological sections showing the expression of ChR2 in the pontine nuclei (PN) and the position of the optical fiber.

**Figure supplement 4.** Histological sections showing targeting of four-shank silicon probes to the deep cerebellar nuclei.

**Figure supplement 5.** Distribution of response latencies to pontine nuclei (PN) stimulation in Purkinje cells, the deep cerebellar nuclei (DCN), and motor cortex.

stimulation with tonic excitation, we also observed cells with different patterns of activity, especially phasic inhibition (*Figure 3A–B*). Furthermore, most Purkinje cells (n = 14/23, Rayleigh test, q < 0.05) were rhythmically entrained to the 40 Hz PN stimulation (*Figure 3—figure supplement 2A–B*). This entrainment, however, was relatively weak: the average across neurons of the mean resultant length was 0.059 (where length 0 corresponds to no entrainment, and length 1 corresponds to perfect entrainment).

We next examined how the effects of PN stimulation propagated into the DCN, which receive inhibitory input from Purkinje cells of the cerebellar cortex, as well as collateral excitatory inputs from the PN (*Figure 1*). We targeted 64-channel silicon probes to the anterior interpositus nucleus during optogenetic stimulation of the PN (*Figure 3C*, upper left) and verified the expression of ChR2 in the PN (*Figure 3—figure supplement 3*, M288-M304) and targeting of the electrode (*Figure 3—figure supplement 4*) in histological sections. The anterior interpositus receives PN input, is known to contain arm-related neurons, and sends ascending projections to motor thalamus, as well as projections to the red nucleus, brainstem, spinal cord (*Sathyamurthy et al., 2020*), and other targets. DCN neurons (n = 139, n = 8 mice) exhibited a wide range of laser-aligned responses, including sustained or transient excitation or inhibition, slow ramps in activity, and combinations of these (*Figure 3C–D*). While some DCN neurons had motifs of transient excitation (*Figure 3C*, neuron 1) or sustained inhibition (*Figure 3C*, neuron 2), which are opposite in sign to the Purkinje cell motifs of transient suppression (*Figure 3A*, neuron 1) and sustained excitation (*Figure 3A*, neuron 2), respectively, the sustained firing rate increases observed in other DCN neurons (bottom rows of *Figure 3D*) might reflect direct excitation from PN collaterals. We next examined rhythmic entrainment of DCN neurons to the 40 Hz PN stimulation, and found that most neurons (n = 92/139, Rayleigh test, q < 0.05) were significantly entrained (*Figure 3—figure supplement 2C*). Furthermore, DCN neurons were much more strongly entrained than Purkinje cells (*Figure 3—figure supplement 2D–E*; cell-averaged mean resultant lengths of 0.224 and 0.059, respectively, rank sum test, p = 4.8e−6), and the peak phase of the DCN population lagged Purkinje cells by 95°, or 6.6 ms (*Figure 3—figure supplement 2D*). This amplification of entrainment is consistent with previous work in brain slices and anesthetized mice demonstrating that even partial synchronization of Purkinje cells to rhythmic stimulation can strongly entrain DCN neurons at the same frequency (*Person and Raman, 2012*). It is also possible that direct excitatory collaterals from the PN might contribute to the strong entrainment in the DCN.

Does the modulation of cerebellar output by PN stimulation propagate into motor cortex, similar to direct stimulation of the cerebellum or the superior cerebellar peduncle (*Holdefer et al., 2000*; *Nashef et al., 2018*; *Nashef et al., 2019*)? To address this question, we again activated PN neurons with ChR2 and recorded neural activity in layer 5 of motor cortex with a 64-channel silicon probe (*Figure 4A*; n = 10 mice). PN stimulation for 2 s evoked neural responses in 107/848 motor cortical neurons, but unlike the responses in Purkinje cells and the DCN, these were relatively stereotypical, mostly consisting of transient firing rate increases (n = 94 neurons, *Figure 4B*, upper left), though a few exhibited transient decreases (n = 13, *Figure 4B*, lower left). Motor cortical responses occurred later than the responses of cerebellar neurons: spike count changes were detected in 5%, 25%, and 50% of modulated cortical cells within 21, 37, and 63 ms of PN stimulation, while the corresponding latencies were 10, 15, and 19 ms for the Purkinje cells and 13, 20, and 25 ms for the DCN, respectively (*Figure 3—figure supplement 5*). These values are likely underestimates of the fastest

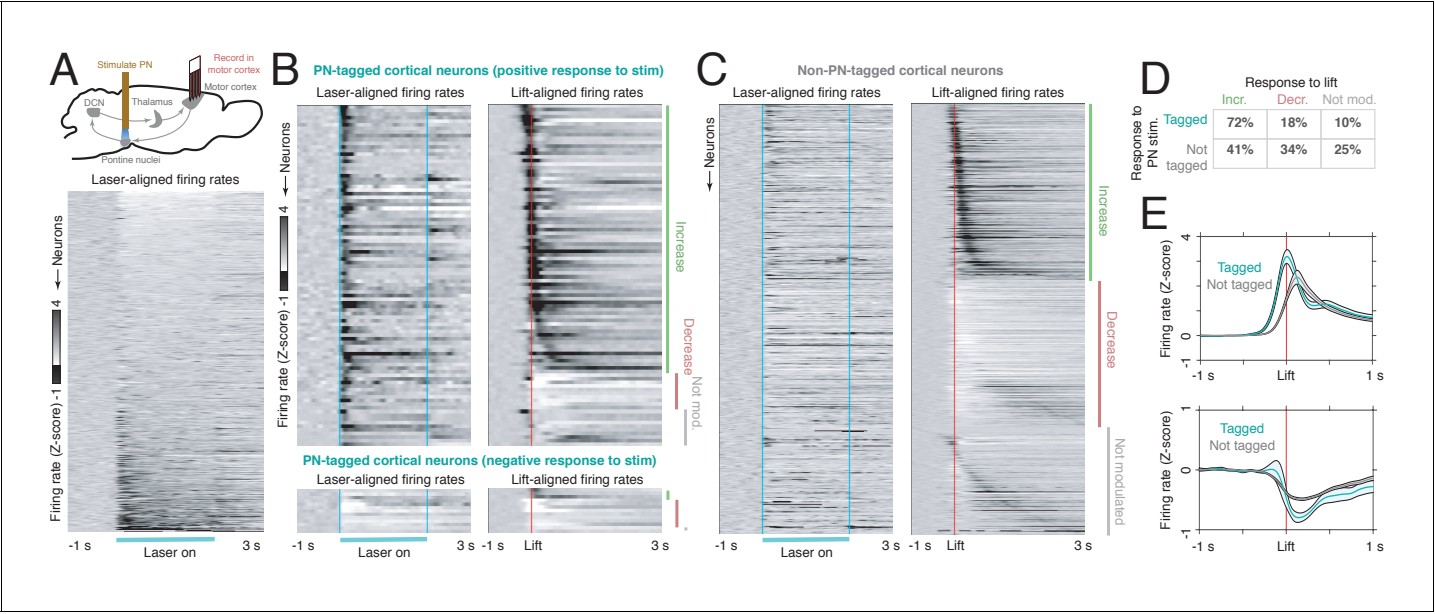

**Figure 4.** Motor cortical neurons receiving feedback from the ponto-cerebellar system have distinct functional properties during reaching. (A) Upper: experimental schematic. Mice expressing ChR2 in the pontine nuclei (PN) were implanted with a tapered optical fiber. A 40 Hz, 2 s laser stimulus was delivered to PN neurons to identify motor cortical cells receiving PN feedback. Then, the activity of PN-tagged and non-PN-tagged neurons was compared during normal reaching. Lower: laser-aligned firing rate z-scores for all cortical neurons (n = 848), sorted by response magnitude. (B) Laser-aligned and movement-aligned neural activity for PN-tagged cortical neurons (n = 107/848, including neurons with laser-aligned increases, n = 94, and decreases, n = 13); n = 26 sessions, n = 8 mice. This includes neurons with both laser-aligned firing rate increases and decreases. Upper panel: neurons with laser-aligned firing rate increases (n = 94). Lower panel: neurons with laser-aligned firing rate decreases (n = 13). Many neurons had increased activity around lift, and a few had decreased activity. (C) Laser-aligned and movement-aligned neural activity for non-PN-tagged cortical neurons (n = 741). Approximately equal numbers of neurons had firing rate increases and decreases around lift. (D) Table showing the percentages of tagged (laser-aligned increases and decreases) and non-tagged neurons with increases, decreases, or no change in activity around lift. (E) Upper: average firing rate z-scores for tagged and non-tagged neurons with firing rate increases around lift. The average activity for the tagged group increased earlier than the activity for the non-tagged group. Lower: average firing rate z-scores for tagged and non-tagged neurons with firing rate decreases around lift.

responses, as they were obtained using sinusoidal, rather than pulse train stimulation. The late-responding neurons are likely not recruited through feedforward excitation, but through unidentified, longer-latency feedback pathways. We next examined whether cortical neurons in which responses were evoked by PN stimulation ('PN-tagged neurons') differed from the rest of the motor cortical population ('non-PN-tagged neurons') in their activity during normal movement. While both tagged and non-tagged neurons exhibited lift-locked increases and decreases, firing rate increases were much more common among tagged neurons (*Figure 4B–D*; chi-square test, p = 1.26e−8). Furthermore, among the cells with firing rate increases, the tagged neurons increased their activity earlier and had a larger peak than the non-tagged neurons (*Figure 4E*, upper), consistent with previous findings in primates using electrical stimulation of the superior cerebellar peduncle (*Nashef et al., 2018*). Thus, during normal movement, the ponto-cerebellar pathway may contribute to the early responses in tagged cortical neurons.

## Optogenetic perturbation of the PN perturbs hand kinematics and impairs reach-to-grab performance

PN stimulation alters activity in the cerebellum and motor cortex, both of which have been implicated in the control of arm movements. What effect does this perturbation have on reaching behavior? In order to address this question, we had animals perform the reaching task, and trials with laser stimulation of the PN were interleaved with control trials (*Figure 5A*, upper). PN stimulation produced a range of effects on reach kinematics in different mice (*Supplementary file 1*). First, in some cases, the reach began normally, but the hand overshot the target, and the animal grabbed at a location past the pellet (*Figure 5A–C*, left and center panels). A difference in the position of the hand at grab was observed in at least one direction in 15/45 sessions from 9/15 mice, and the

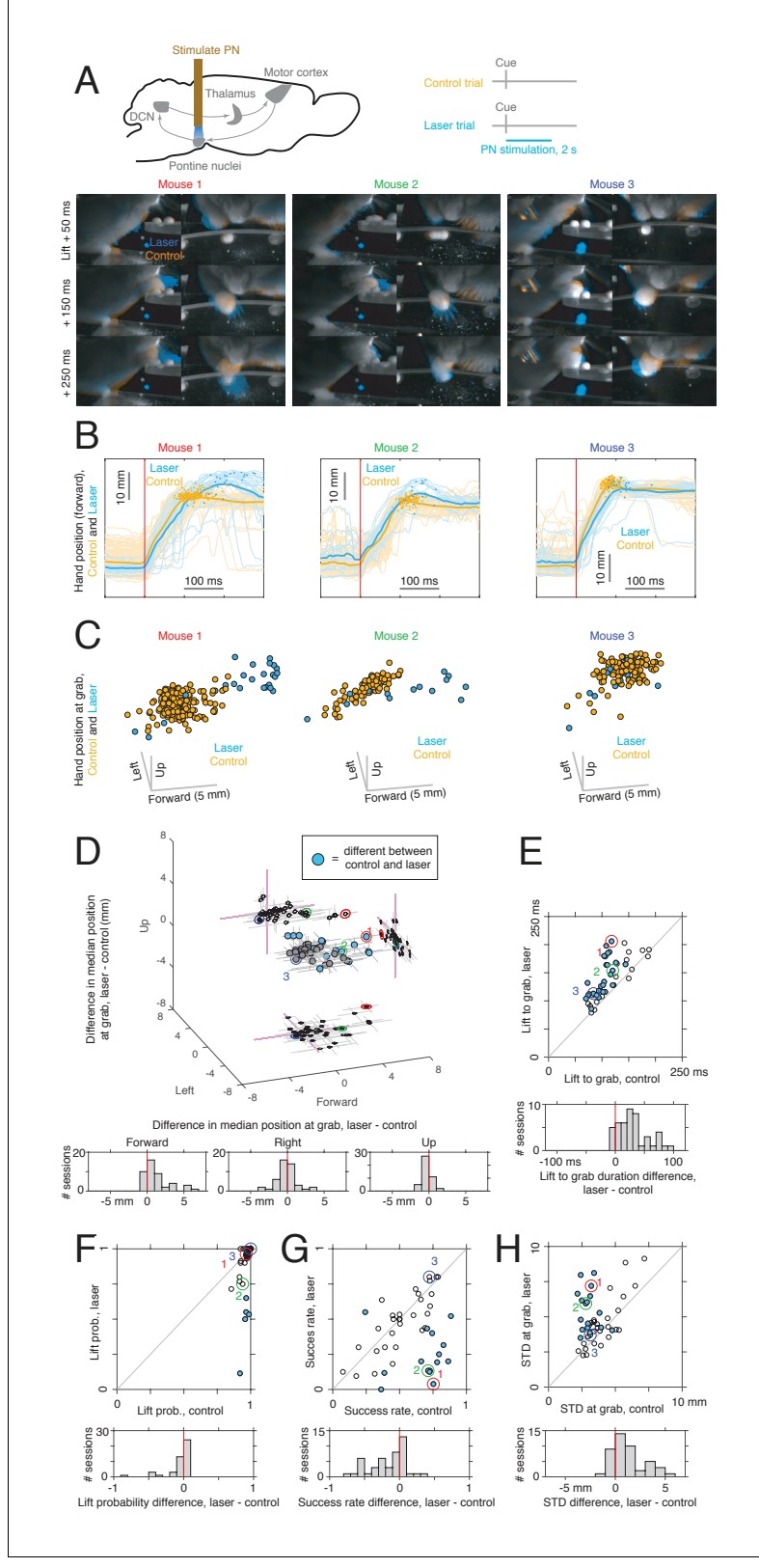

**Figure 5.** Effects of pontine nuclei (PN) stimulation on reaching. (**A**) Upper: experimental schematic. Lower: video frames for control and PN laser stimulation trials for sessions from three mice. Frames from the front and side cameras are shown at offsets of 50, 150, and 250 ms from lift, averaged across trials (see Materials and methods). Orange indicates higher image intensity on control trials, and blue indicates higher intensity on laser trials. For the

*Figure 5 continued on next page*

*Figure 5 continued*

first two mice (left panel), this visualization shows that the animals reach farther forward on laser trials than control trials. Panels labeled 1, 2, and 3 correspond to M270 (8-15-18), M282 (12-5-18), and M240 (9-22-17), respectively. (B) Hand trajectories for the three example mice in (A). Dots indicate the position of the hand at grab. (C) Three-dimensional position of the hand at the time of grab on control and laser trials for each of the three mice in (A–B). Each point corresponds to a single trial. (D) Upper: difference in median hand position at grab between laser and control trials. Each point corresponds to a single experimental session (n = 45). Filled blue dots correspond to sessions in which a difference between control and laser was found in at least one direction (two-sided rank sum test for each direction, q < 0.05). Lines indicate bootstrapped 95% confidence intervals. Lower: histogram of differences in hand position in each spatial dimension between laser and control conditions. (E) Upper: median delay from lift to grab, laser vs. control; filled dots indicate sessions where a difference between laser and control was detected (two-sided rank sum test, q < 0.05). Lower: histogram of difference between lift-to-grab duration on laser and control trials. (F) Upper: probability of initiating a reach, laser vs. control; filled dots represent sessions where a difference between control and laser was detected (chi-square test, q < 0.05). Lower: histogram of difference in reach probability on laser and control trials. (G) Upper: success rate on first reach attempt for laser vs. control; filled dots indicate a difference between laser and control (chi-square test, q < 0.05). Lower: histogram of differences between success rate on laser and control trials. (H) Upper: standard deviation (summed across forward, left, and up directions) in the position of the hand at grab for laser vs. control trials. Each point corresponds to one experimental session, and filled dots represent sessions where a difference between control and laser was detected in at least one direction (two-sample F-test for equal variances, two-sided, q < 0.05). Lower: histogram of differences in grab position standard deviation between laser and control trials. Note: neural activity in the motor cortex, cerebellar nuclei, or both was also recorded in these sessions.

largest differences occurred in the forward direction, corresponding to overreaching (*Figure 5D*; *Video 1*). Second, the time from lift to grab increased relative to control (*Figure 5E*); this occurred in 27/45 sessions from 11/15 mice. In some sessions, this was not due to overreaching the target, but to a reduction in hand velocity (*Figure 5A–C*, right panels). Third, in a few cases (5/45 sessions from 4/15 mice), stimulation reduced the probability of movement initiation (*Figure 5F*, *Video 2*). Fourth, stimulation reduced the rate of success on the first reach attempt in 14/45 sessions from 8/15 mice (*Figure 5G*; one session showed the opposite effect). Fifth, the dispersion in the position of the hand at grab was higher on laser trials in 17/45 sessions from 8/15 mice. At least one effect was observed in 32/45 sessions from 14/15 mice. These results demonstrate that PN stimulation impairs reaching performance, typically by disrupting precision, accuracy, duration, or success rate of the movement.

## Effects of PN stimulation on the cerebellar nuclei and motor cortex during reaching

How does PN stimulation affect neural dynamics in the DCN and motor cortex during reaching? We recorded from the DCN on control trials, in which the animal reached to the pellet following a cue, and laser trials, when PN stimulation was delivered throughout the trial (*Figure 6A*, lower inset; n = 139 neurons from n = 12 sessions in n = 8 mice). On control trials, DCN neurons were strongly modulated around movement (*Figure 6A*, orange

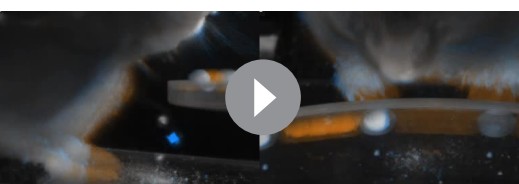

**Video 1.** Effects of pontine nuclei (PN) stimulation in an example mouse (animal 1 in *Figure 5*; M270, 8-15-18). Raw video from all trials in the session are overlaid, with control trials in orange and PN stimulation trials in blue. Data are aligned to movement onset (see Materials and methods).
https://elifesciences.org/articles/65906#video1

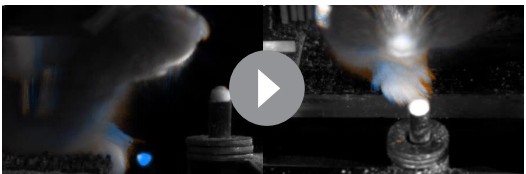

**Video 2.** Effects of pontine nuclei (PN) stimulation in an animal in which reach initiation was robustly blocked (M295, 9-9-19). Raw video from all trials in the session are overlaid, with control trials in orange and PN stimulation trials in blue. Data are aligned to the start of the trial.
https://elifesciences.org/articles/65906#video2

traces; *Figure 6B*; n = 60 neurons with firing rate increases, and n = 70 neurons with firing rate decreases), consistent with previous results in primates (*Fortier et al., 1989*; *Fortier et al., 1993*) and mice (*Becker and Person, 2019*). Lift-aligned activity on laser trials was typically similar to the control pattern (*Figure 6A*, left and center), but in some cells, the stimulation abolished the normal pattern of activity (*Figure 6A*, right). In order to quantify the similarity between neural activity on control and laser trials, we first computed the correlation between the firing rates of all neurons on control trials and all neurons on laser trials at each peri-lift time point. This correlation became positive shortly before movement onset and remained high throughout the movement: the cells that had higher firing rates at a given time on control trials also tended to have higher firing rates on laser trials at that time (*Figure 6C*, upper). Next, for each neuron, we computed the correlation between firing rates on control and laser trials over the entire peri-lift window; this provides an estimate of how similar the control and laser patterns are for individual neurons. These correlations tended to be positive and large (positive for 131/139 neurons, negative for 5/139 neurons; permutation test with large-sample approximation, q < 0.05; median correlation of 0.62). Nonetheless, we did observe laser-induced firing rate changes during the movement (*Figure 6D*), which were correlated with the changes predicted by neurons' response to the laser alone (*Figure 6—figure supplement 1A–E*), except in a short period around movement onset (*Figure 6—figure supplement 1E*, upper) when sensory feedback may also play a larger role. That is, firing rate changes induced by PN stimulation during movement were consistent with the changes expected based on neurons' separate responses to the laser and reach alone. Thus, for most DCN neurons, PN stimulation did not severely disrupt activity patterns during reaching, but instead subtly modulated these patterns (*Figure 6B–D*).

While the modulation of DCN activity by PN stimulation might perturb reach kinematics through descending routes to the spinal cord (*Figure 1B*), cerebellar output is also known to alter activity in motor cortex during behavior (*Meyer-Lohmann et al., 1975*; *Nashef et al., 2019*). To test the extent to which PN stimulation influenced motor cortical activity during movement, we recorded from motor cortical ensembles (n = 1157 neurons from n = 38 sessions in n = 10 mice) during reaching on trials with PN stimulation and on control trials (*Figure 7A*, lower inset). Similarly to DCN neurons, cells in motor cortex were strongly modulated during movement on control trials and often exhibited small differences on laser trials (*Figure 7A*). Firing rates on control and laser trials were positively correlated over time for most cells (*Figure 7B–C*; positive correlation for 965 neurons and negative correlation for 47 neurons; permutation test with large-sample approximation, q < 0.05; median correlation 0.67). However, as with DCN activity, cortical activity during the task was modulated by PN stimulation (*Figure 7D*). Consistent with the observation that cortical responses to laser-only stimulation were mainly transient increases (*Figure 7—figure supplement 1A*), the effect of PN stimulation during the task was correlated with the effect predicted from the laser-only responses immediately before movement onset (*Figure 7—figure supplement 1B–E*). Later differences between activity on control and laser trials could not be accounted for by laser-only responses (*Figure 7—figure supplement 1D–E*), but might reflect differences in behavior. The difference between lift-aligned activity on laser and control trials was larger for cortical neurons tagged by PN stimulation alone (i.e., the neurons in *Figure 4B*), especially before the initiation of the reach, near the onset of PN stimulation (*Figure 7—figure supplement 2*).

How similar is population activity in the DCN and motor cortex during reaching at the level of single trials? To address this question, in the sessions with simultaneous cortical and DCN recordings (n = 5), we performed canonical correlation analysis (CCA) between the neural activity recorded in the two regions (*Johnson and Wichern, 2007*). For this analysis, we considered the first four principal components of multiunit activity recorded in a 400 ms window around movement (starting 100 ms before lift), during all control and PN stimulation trials (see Materials and methods). CCA was used to find pairs of canonical variates, that is, linear transformations of the original principal components whose trajectories (scores) were maximally correlated between the two regions (*Figure 8A*). In each session, CCA found two pairs of canonical variates that were strongly correlated (scores with correlations in the range 0.58–0.88, *Figure 8B*) and together accounted for half or more of the variance in each region (49–64% in cortex, 51–70% in DCN; *Figure 8C*, left column). Overall, between a quarter and a half of the variance in each region was accounted for by canonical variates in the other region (25–50% of variance in DCN explained by cortex, 26–47% in cortex explained by DCN; *Figure 8C*, right column). Thus, within the constraints of our behavioral task, some but not all of the dominant dimensions of cortical and cerebellar activity are strongly correlated.

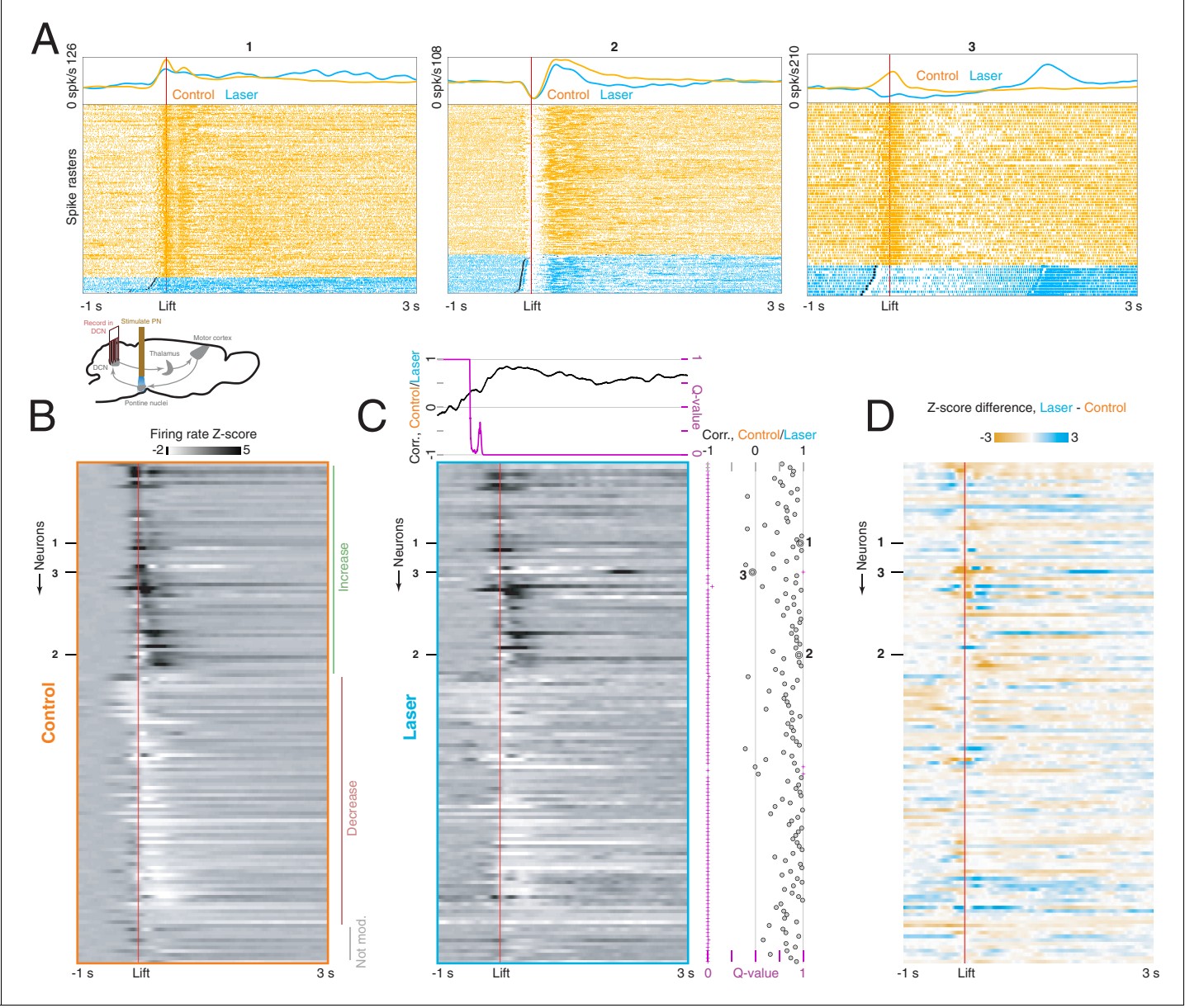

**Figure 6.** Effect of pontine nuclei (PN) stimulation on neural activity in the deep cerebellar nuclei (DCN) during reaching. (A) Lower inset: experimental schematic. Animals performed the reaching task as neural activity in the DCN was recorded. On some trials, PN neurons were activated with a 40 Hz, 2 s laser stimulus. Top panels: lift-aligned spike times and average firing rates on control (orange) and laser trials (blue) for three DCN neurons. Black dots indicate the laser onset. (B) Heatmap displaying lift-aligned average firing rate z-scores on control trials for all DCN neurons (n = 139 neurons, n = 8 mice). (C) Heatmap displaying lift-aligned average firing rate z-scores for laser trials. Upper inset: the black line shows the rank correlation (Spearman's ρ) between firing rates of the 139 neurons on control and laser trials at each time point. The magenta inset shows the q-value against the null hypothesis that the correlation is zero. Right inset: black circles show the rank correlation (Spearman's ρ) between the control and laser firing rates over time for each neuron. Magenta crosses show the q-value against the null hypothesis that the correlation of control and laser values over time is zero. (D) Difference between lift-aligned z-scores for control and laser trials. Orange regions indicate neurons and time points in which the firing rate is higher on control trials, and blue regions indicate points in which the firing rate is higher on laser trials.

The online version of this article includes the following figure supplement(s) for figure 6:

**Figure supplement 1.** Relationship between laser-aligned and lift-aligned activity in the deep cerebellar nuclei (DCN).

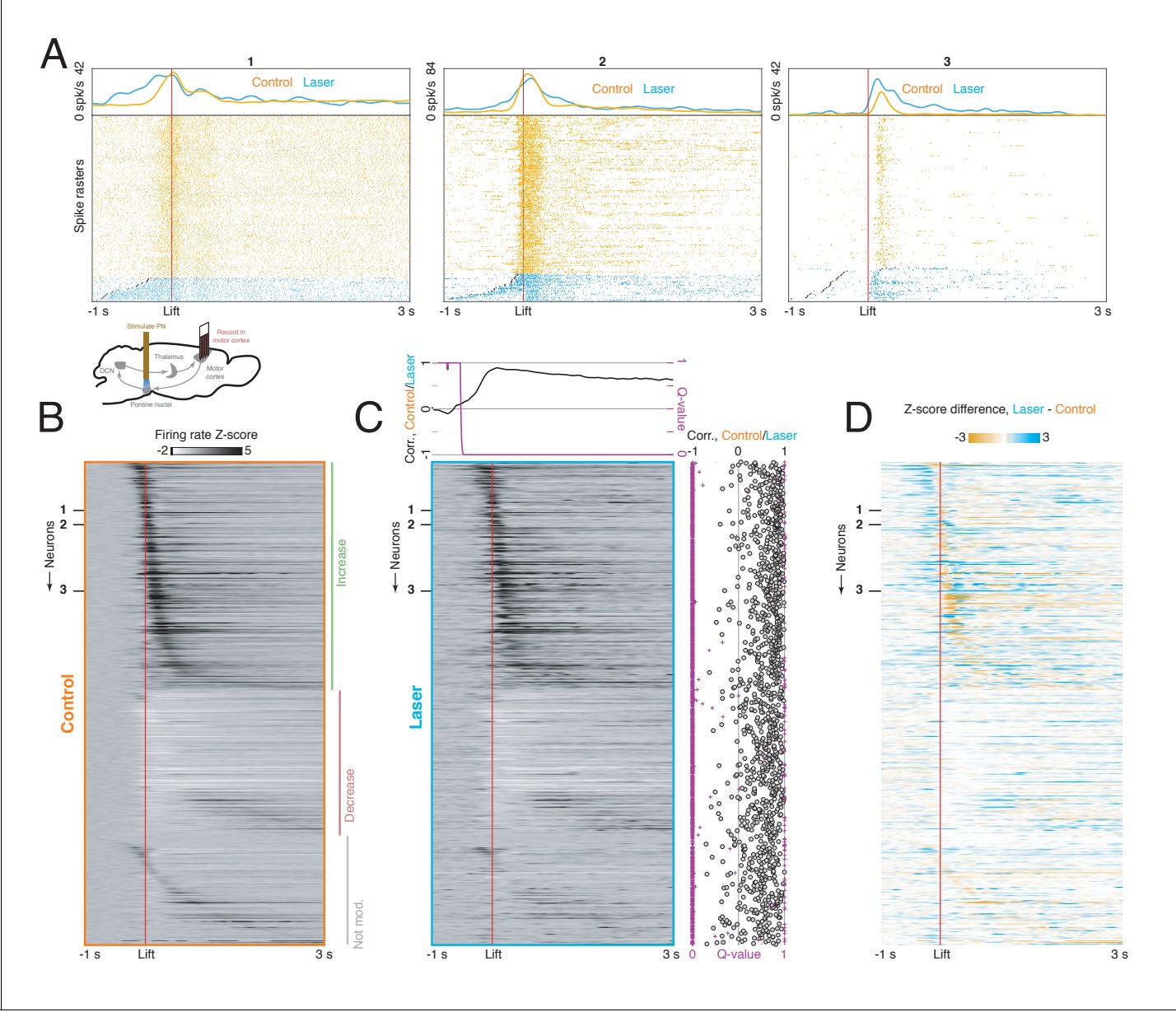

**Figure 7.** Effect of pontine nuclei (PN) stimulation on neural activity in motor cortex during reaching. (**A**) Lower inset: experimental schematic. Animals performed the reaching task as neural activity in motor cortex was recorded. On some trials, PN neurons were activated with a 40 Hz, 2 s laser stimulus. Top panels: lift-aligned spike times and average firing rates on control and laser trials for three motor cortex neurons. Black dots indicate the laser onset. (**B**) Heatmap displaying lift-aligned average firing rate z-scores on control trials for all motor cortex neurons (n = 1157 neurons, n = 10 mice). (**C**) Heatmap displaying lift-aligned average firing rate z-scores for laser trials. Upper inset: the black line shows the rank correlation (Spearman's ρ) between firing rates of the 1157 neurons on control and laser trials at each time point. The magenta inset shows the q-value against the null hypothesis that the correlation is zero. Right inset: black circles show the rank correlation (Spearman's ρ) between the control and laser firing rates over time for each neuron. Magenta crosses show the q-value against the null hypothesis that the correlation of control and laser values over time is zero. (**D**) Difference between lift-aligned z-scores for control and laser trials. Orange regions indicate neurons and time points in which the firing rate is higher on control trials, and blue regions indicate points in which the firing rate is higher on laser trials.

The online version of this article includes the following figure supplement(s) for figure 7:

**Figure supplement 1.** Relationship between laser-aligned and lift-aligned activity in motor cortex.

**Figure supplement 2.** Difference in lift-aligned cortical activity on pontine nuclei (PN) stimulation and control trials for PN-tagged and non-tagged neurons.

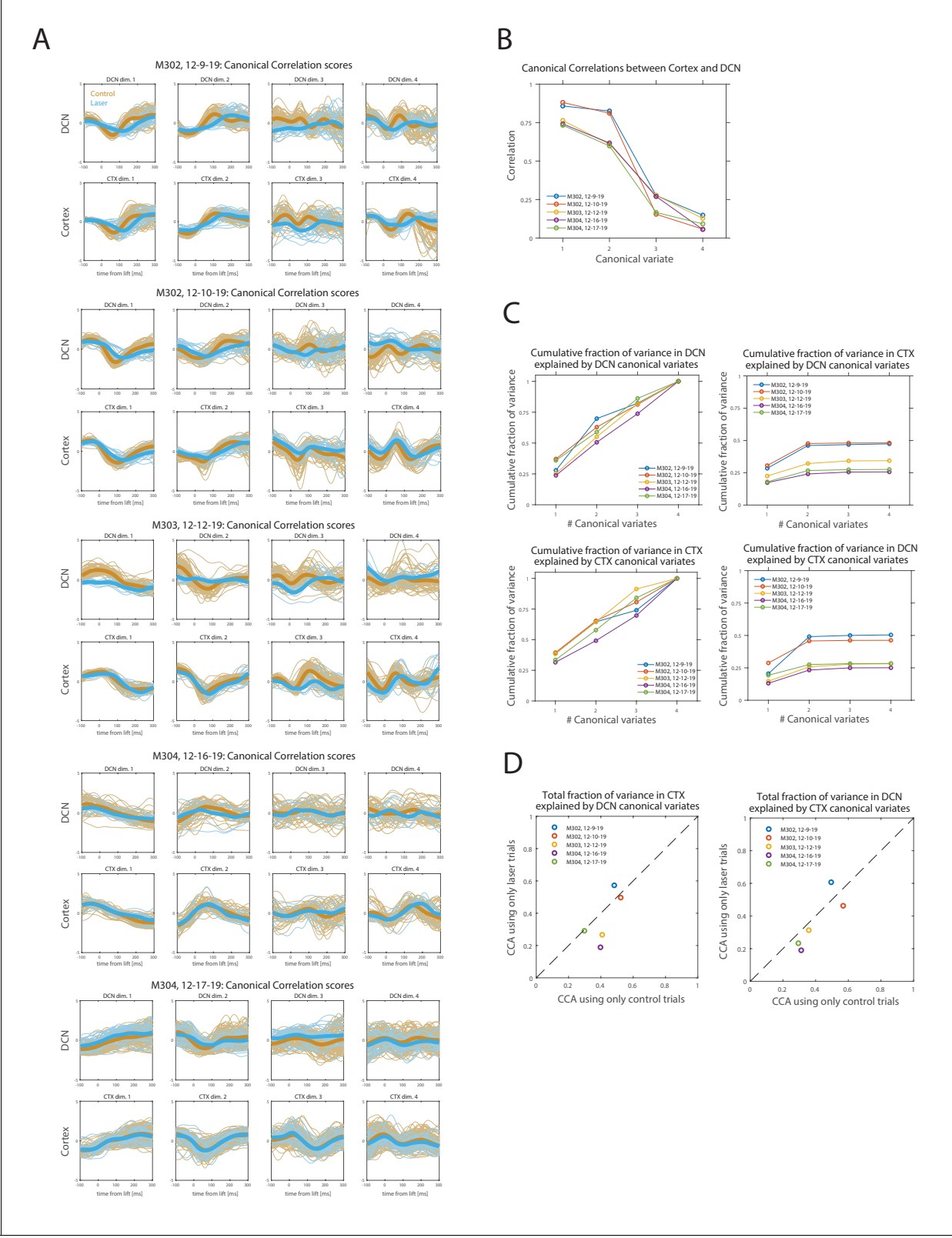

**Figure 8.** Canonical correlation analysis (CCA) of simultaneously recorded population activity (first four principal components) in motor cortex (CTX) and deep cerebellar nuclei (DCN). (**A**) Lift-aligned canonical correlation scores for CTX and DCN for each session (n = 5; see Materials and methods). Each column is a pair of canonical variates, ordered by canonical correlation values. Control trials are orange and pontine nuclei (PN) stimulation (laser) trials are blue, with thicker lines representing the means. Trials of both types were used in CCA. (**B**) Canonical correlations for the four pairs of canonical

*Figure 8 continued on next page*

*Figure 8 continued*

variates, in each of the sessions in (**A**). (**C**) Cumulative fraction of (standardized) sample variance in the population activity recorded in CTX or DCN that is explained by an increasing number of canonical variates in the same or opposite region. The total fraction of variance explained by the complete set of canonical variates (#canonical variates = 4) is one by construction within the same region but depends on canonical correlations for the opposite region. (**D**) Comparison of the total fraction of variance explained in CTX (left) or DCN (right) by the complete set of canonical variates of the opposite region, between two alternative CCAs performed using either only laser trials or only control trials (matching the number of laser trials, see Materials and methods).

To assess the effect of the PN perturbation on the similarity of population activity in the two regions, we compared the results of two separate CCAs, one using only activity in laser trials and the other using only control trials (matching the number of laser trials for each session; Materials and methods). The total fraction of variance in one region explained by the other region was usually lower for CCA using laser trials than for CCA using control trials, but the differences were small (*Figure 8D*). This suggests that PN perturbation has a limited effect on the similarity of population activity in cortex and DCN during reaching.

## Decoding of hand velocity from neural activity on control and PN stimulation trials

For both the DCN and cortex, we observed differences in neural activity between control and PN stimulation (laser) trials during the movement. Could these neural differences explain the behavioral differences? To address this question, we first designed linear filters to decode 3D hand velocity on control trials from principal components of multiunit activity in motor cortex and DCN (n = 38 sessions with cortical recordings, n = 10 sessions with DCN recordings; Materials and methods). We assessed the performance of these decoders by computing the goodness-of-fit ($R^2$) between observed and decoded hand velocities on a test set of control trials not used for training the decoder (*Figure 9A* and *Figure 9—figure supplement 1A–B*, orange lines). We found that cortical and DCN decoders were comparable in their performance (mean $R^2 = 0.55$ and $R^2 = 0.54$, respectively; *Figure 9—figure supplement 1C–D*, x-axis). In the sessions in which motor cortex and DCN were recorded simultaneously (n = 5), we trained a decoder on pooled activity from both regions and compared its performance with the cortical decoder. We found that the addition of DCN activity to the cortical decoder improved the performance very marginally (mean $R^2 = 0.78$ vs. $R^2 = 0.77$; *Figure 9—figure supplement 1E*, Materials and methods) suggesting that the two populations carry similar information about hand velocity.

We then applied the velocity decoders, that were trained only on control trials, to the neural activity in laser trials (*Figure 9A* and *Figure 9—figure supplement 1A–B*, blue lines; *Figure 9—figure supplement 1C–D*, y-axis; *Figure 9—figure supplement 1F*). Decoded velocity differed between control and laser trials (*Figure 9B–C*, center columns), confirming that perturbation of the PN altered dimensions of cortical and DCN activity related to arm kinematics. We next examined whether the differences between hand velocity on control and laser trials predicted by the neural decoders were consistent with the observed velocity differences. Typically, the decoded velocity differences qualitatively matched the observed differences (*Figure 9A*, left vs. center and right columns; *Figure 9B–C*, left vs. center columns; *Video 3*). Across experimental sessions, the correlations between the decoded and observed velocity differences were positive in each direction for the cortical decoder (*Figure 9B*, right columns) and in the lateral direction for the DCN decoder (*Figure 9C*, right columns). The goodness-of-fit between the decoded and observed velocities was lower for laser trials than for control test trials, especially for DCN (mean $R^2 = 0.30$ in laser trials vs. $R^2 = 0.54$ in control trials for DCN; mean $R^2 = 0.42$ vs. $R^2 = 0.55$ for cortex; *Figure 9—figure supplement 1C–D*). This drop in decoder performance on laser trials might reflect an altered mapping between neural activity and hand velocity, but might also reflect generalization error, since the decoders were trained only on control trials. Consistent with the latter interpretation, an alternative procedure in which the decoder was trained on a balanced set of control and laser trials yielded similar performance on both trial types (mean $R^2 = 0.41$ in laser trials vs. $R^2 = 0.43$ in control trials for DCN; mean $R^2 = 0.47$ vs. $R^2 = 0.45$ for cortex; *Figure 9—figure supplement 1G–H*; Materials and methods). Overall, the decoding results show that the changes in cortical and DCN trajectories induced by PN stimulation during movement parallel the observed changes in hand velocity. Thus, disrupting

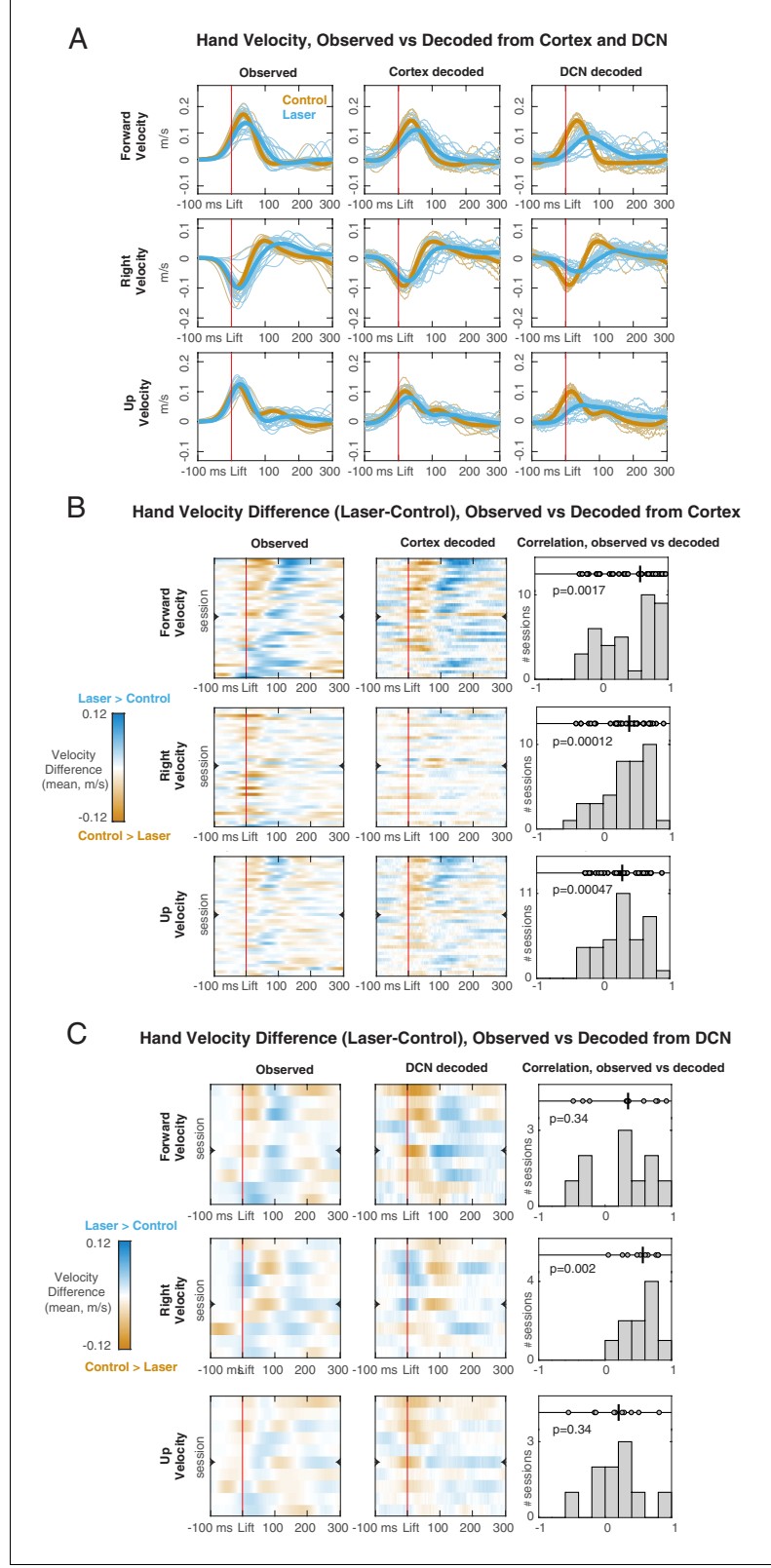

**Figure 9.** Hand velocity decoding from neural activity in motor cortex and deep cerebellar nuclei (DCN). (**A**) Observed (left column) vs. decoded (center cortex, right DCN) velocity trajectories in each direction for a session with simultaneous recordings. Blue lines represent all trials with pontine nuclei (PN) perturbation ('laser') and their mean (thicker line). Orange lines represent all 'test control' trials, not used for training the decoder, and their

*Figure 9 continued on next page*

*Figure 9 continued*

mean. (B) Observed vs. decoded mean differences in velocity between laser and control test trials, for each of the sessions with cortical recordings (n = 38). In the heatmaps (left observed, center decoded) rows are sessions, ordered based on the observed mean velocity difference in the forward direction in the 100 ms following lift. Orange regions indicate time points in which the velocity is higher on control trials, and blue regions indicate points in which the velocity is higher on laser trials. The row denoted with a mark is the session shown in (A). Right panels show the distribution of rank correlations (Spearman's ρ) between the observed and decoded differences in velocity over time (i.e., between corresponding rows of the heatmaps). Within each panel, individual correlations and their median (black line) are shown on top, with p-value of the two-sided sign rank test against the null hypothesis of zero median, and histogram of correlation values across sessions is shown at the bottom. (C) Same as (B), for each of the sessions with DCN recordings (n=10).

The online version of this article includes the following figure supplement(s) for figure 9:

**Figure supplement 1.** Hand velocity decoding from neural activity in motor cortex (CTX) and deep cerebellar nuclei (DCN), for all sessions with recordings in at least one of the regions (n=38 CTX sessions, n=10 DCN sessions; n=5 sessions had simultaneous recordings in both regions).

cortico-cerebellar communication redirects cerebellar and cortical population trajectories to generate movements with different kinematic features.

## Discussion

Previous work has demonstrated that the total output of motor cortex is critical for driving reaching movements: silencing excitatory neurons in the motor cortex of mice robustly blocks the initiation of reaching and interrupts the execution of ongoing movement (*Guo et al., 2015*; *Sauerbrei et al., 2020*; *Galiñanes et al., 2018*; *Castro, 1972*). This output, however, is routed to many different targets, which are broadly distributed across the spinal cord, brainstem, basal ganglia, thalamus, cerebral cortex, and – through the PN – the cerebellum. It has been challenging to disentangle the function of each of these output channels, because individual pyramidal tract neurons frequently project to several targets (*Kita and Kita, 2012*; *Winnubst et al., 2019*). For example, because many PN-projecting cortical neurons also project to the spinal cord, optogenetically perturbing these neurons would disrupt both cortico-cerebellar and corticospinal communication, and the effects of the perturbation on behavior could not be reliably attributed to impairment of the cortico-cerebellar pathway. Furthermore, by demonstrating that perturbation of a single source of mossy fiber input (which overlaps with inputs from sensory and other sources) impairs reaching, our work goes beyond previous studies documenting impairment from direct manipulation of the cerebellum itself (*Brooks et al., 1973*; *Conrad et al., 1975*; *Meyer-Lohmann et al., 1975*; *Conrad and Brooks, 1974*; *Becker and Person, 2019*).

In the present study, we take advantage of an anatomical bottleneck in the PN, which receives extensive cortical input and projects selectively to the cerebellum. By perturbing the PN, we selectively disrupt communication between the forebrain and cerebellum, and are able to examine how this disruption alters behavior and neural dynamics across the cortico-cerebellar loop. In contrast with direct manipulations of cortex, our perturbation of the PN did not typically block movement initiation. Instead, this perturbation disrupted the skill and precision of the movement, inducing a decrease in success rate, an increase in the variability of the endpoint, or an increase in movement duration. Furthermore, the perturbation altered reach-aligned neural activity in the DCN and motor cortex, and a decoding analysis revealed that these changes in neural activity predicted the changes in arm kinematics. We interpret these results as evidence that while the total output of motor

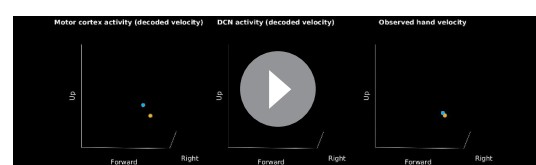

**Video 3.** Decoded and observed hand velocities for an example session (M302, 12-9-19). Right: average hand velocity observed within −100 to 300 ms of movement onset. Orange corresponds to control trials, and blue to PN stimulation trials. Left and center: decoded hand velocity from motor cortex and DCN, respectively, in the same session.

https://elifesciences.org/articles/65906#video3

cortex drives reaching (*Guo et al., 2015*; *Sauerbrei et al., 2020*; *Galiñanes et al., 2018*), the cortico-cerebellar loop selectively contributes to small adjustments to the movement – that is, to fine-tuning it.

We stimulated the ponto-cerebellar pathway in the absence of reaching to characterize the signal transformation properties of the cerebello-cortical loop. In Purkinje cells, optogenetic stimulation of the PN evoked sustained firing rate increases, likely driven by the excitatory granule cell pathway, and transient firing rate decreases, which could reflect feedforward inhibition from molecular layer interneurons, but not sustained decreases. Neurons with long onset latencies (e.g., greater than 30 ms) are likely recruited through recurrent loops in the network, but our experiments do not enable us to identify these loops. One synapse downstream in the DCN, PN stimulation evoked complementary motifs of transient excitation and sustained inhibition for some neurons, consistent with the expected effects of inhibition from Purkinje cells. Other DCN neurons, however, exhibited sustained firing rate increases, which could reflect the effect of excitatory collaterals from the PN. We also observed that PN stimulation altered cerebellar activity beyond changes in mean firing rate. Most Purkinje cells and DCN neurons were significantly entrained to the 40 Hz stimulation, and this entrainment was much stronger in the DCN. This observation suggests that the corticonuclear projection may amplify periodic inputs, consistent with previous findings in brain slices and anesthetized mice (*Person and Raman, 2012*). In motor cortex, responses to PN stimulation differed from those in the DCN in several key respects. First, responses were observed in fewer cortical than cerebellar neurons, suggesting that the cerebello-cortical pathway attenuates PN signals, rather than amplifying them. Second, cortical responses were more transient than those in the DCN, suggesting that the cerebello-cortical pathway can filter out steady-state firing rate changes in the DCN. Third, pontine stimulation mainly produced increases in cortical firing rates, in contrast with the diverse responses in the DCN (*Figure 6—figure supplement 1A* vs. *Figure 7—figure supplement 1A*). Overall, these results suggest that the cerebellum provides temporally filtered feedback to selectively modulate a subpopulation of cortical neurons.

Unlike damage to or inactivation of motor cortex (*Guo et al., 2015*; *Miri et al., 2017*; *Galiñanes et al., 2018*; *Sauerbrei et al., 2020*; *Krakauer and Thomas Carmichael, 2017*), stimulation of the PN in a cued reaching task did not usually prevent initiation of the movement, though this was observed in a few cases (blue points below the diagonal in *Figure 5F*; *Video 2*). Instead, PN stimulation impaired motor skill by altering a range of kinematic features, including reach-to-grasp time, endpoint position, and endpoint variance. This range of effects resembles the dysmetria observed in spinocerebellar ataxia patients but does not replicate other deficits, such as intention tremor. It is possible that the variability of behavioral effects could be due, in part, to subtle differences in pontine ChR2 expression, the placement of the fiber, light power delivered to the tissue, or behavioral strategies adopted by individual mice. It is also possible that the cerebellum may rapidly adapt to the PN perturbation across trials, and that different animals may adapt to different degrees. Furthermore, we cannot exclude the possibility that a different manipulation, such as higher-power stimulation, might have resulted in a robust block of movement initiation. Although our perturbation only disrupted one channel of cerebellar inputs, its effects on behavior were consistent with those obtained in previous studies by manipulation of cerebellar output. In primates, high-frequency electrical stimulation of the superior cerebellar peduncle increased the tortuosity and variability of reaches and decreased their speed (*Nashef et al., 2019*), consistent with our findings that PN stimulation increased endpoint variance and movement duration. These parallel results held even though Nashef et al. disrupted cerebellar projections to the thalamus, and we disrupted a single mossy fiber system. Previous work in cats has demonstrated that pharmacological inactivation of the interpositus nucleus increases the variability of reaching movements and leads to mistargeting of the paw (*Martin et al., 2000*). In mice, optogenetic stimulation of the interpositus nucleus decreases outward reach velocity and results in a hypometric endpoints, while silencing the same region increases the outward velocity and leads to hypermetric endpoints (*Becker and Person, 2019*), similar to the effect of ablating a glutamatergic subpopulation (*Low et al., 2018*). This is consistent with our observation that PN stimulation altered the duration of the movement and, often, induced an overshoot of the target. Consistent with these previous studies, our findings support the idea that the cortico-cerebellar loop is necessary for fine-tuning, rather than driving, the arm kinematics during skilled reaching.

Neural activity during reaches with PN stimulation largely recapitulated the patterns observed during normal reaching. Why might the neural dynamics in each area, as well as the correlations between them, be robust to disruption of one of the largest inputs to the intermediate and lateral cerebellum? Both DCN and cortex receive inputs from multiple pathways that do not include PN. Since these other pathways are not directly influenced by our perturbation of the PN, it is possible that they override the aberrant PN activity and entrain relatively normal activity in the DCN and cortex. For DCN, potential inputs include sensory streams, such as those from external cuneate nucleus (*Huang et al., 2013*), or motor inputs, such as those from the red nucleus (*Beitzel et al., 2017*). These inputs may act by directly influencing the DCN or by driving near normal Purkinje cell output from the cerebellar regions that they target. Our results suggest that future work will need to consider how dynamics in the cortico-cerebellar system integrate with those of other loops between the forebrain, hindbrain, spinal cord, and musculoskeletal plant to enable skilled movement.

## Materials and methods

### Transgenic mouse lines

Slc17a7-Cre (VGLUT1-Cre) mice were generated by the Janelia Research Campus Gene Targeting and Transgenics Facility (*Huang et al., 2013*). For the PN stimulation experiments, we used Slc17a7-Cre mice (n = 15 for behavior and cortex/DCN electrophysiology, n = 4 for cerebellar cortex electrophysiology). For the electrophysiological recordings in the PN, we obtained mice (n = 6) expressing ChR2 in pyramidal tract neurons by crossing the Cre driver line (Tg(Sim1-Cre)KJ18Gsat, The Jackson Laboratory) to a Cre-dependent ChR2 reporter mouse, Ai32 (Rosa-CAG-LSL-ChR2(H134R)-EYFP-WPRE, The Jackson Laboratory). No statistical methods were used to determine the number of animals used. Numbers of mice were within the standard range for systems neuroscience studies. Where possible, statistical comparisons were performed over trials within each session, rather than over animals, enabling us to replicate findings across sessions and characterize animal-to-animal variability.

### Stereotaxic surgeries

All mice (2–15 months of age) were anesthetized with 2% isoflurane and placed on a heating pad in a stereotactic frame (Kopf Instruments, Tujunga, CA). Mice were administered 0.1 mg/kg of subcutaneous buprenorphine at the time of surgery, and 5 mg/kg of subcutaneous ketoprofen at the end of surgery, followed by an additional dose once a day for 2 days. The scalp was sanitized with three alternating rounds of iodine surgical scrub and alcohol, a portion of the scalp was removed, the skull was cleaned, and a custom-made headpost was affixed with UV-curing cement (OptiBond, Kerr; Calibra Universal Self-Adhesive Resin, Dentsply Sirona) or dental acrylic (RelyX Unicem, 3M). For viral injections into the PN, craniotomies were made using a dental drill over the PN (3.90 mm posterior to bregma, 0.4 mm lateral). Injections were made on the left (contra-limb) side. Injection pipettes were made from glass capillaries pulled on a Sutter P-97 (Sutter, Novato, CA). Viruses were loaded using a Narishige pneumatic injector (MO-10, Tokyo, Japan) and injected into the PN at 5.9, 5.6, and 5.3 mm below the dural surface. Injection volumes were 100 nl each. After each injection, at least 2 min elapsed before proceeding to the next depth, and the pipette was withdrawn 5 min after the final injection. For the ChR2 PN stimulation experiments, Slc17a7-Cre animals (n = 15 for behavior and cortex/DCN electrophysiology, n = 2 for cerebellar cortex electrophysiology) were injected with AAV-2/1-CAG-flex-ChR2-TdTomato and implanted with tapered optical fibers (*Pisanello et al., 2017*) in the left PN (Optogenix, Lecce, Italy; NA 0.39, taper angle 6.9°, active length ~1 mm, core diameter 200 μm, core+cladding 225 μm). The pattern of ChR2 expression and the placement of the fiber were assessed in postmortem histology (*Figure 3—figure supplement 3*). For the PN optogenetic perturbation experiments, a craniotomy targeting forelimb motor cortex (bregma +0.5, left 1.7 mm), the DCN (bregma −6.3, right 1.8 mm), or the cerebellar cortex (bregma −7.0, right 2.5 mm) was sealed with silicone elastomer (Kwik-Sil, WPI). Craniotomies in motor cortex and the cerebellar nuclei were performed in ten and eight mice, respectively, with three of these mice receiving craniotomies in both regions. Recordings in the cerebellar cortex were performed in four mice (two for multiunit depth profile recording and two for single-unit Purkinje cell recording). Following surgery, injections of ketoprofen (5 mg/kg) and buprenorphine (0.1 mg/kg; Henry Schein Animal Health,

Melville, NY) were administered subcutaneously. If animals exhibited signs of pain or distress following surgery, additional doses of either ketoprofen or buprenorphine were administered, as directed by veterinary staff. All procedures were performed in accordance with protocols approved by the Institutional Animal Care and Use Committee (IACUC) of the Janelia Research Campus (protocols 16–139 and 19–177).

## Histology

Mice were anesthetized and euthanized with isoflurane until no breathing was visually detected for ~30 s. Mice were then transcardially perfused with 1× PBS (Fisher Scientific Inc, 20 ml), followed by 4% paraformaldehyde (formalin, 20 ml, Fisher Scientific Inc). Brain tissue was removed, incubated in formalin for 24–72 hr, then rinsed and stored in 1× PBS. Brains were sectioned at 70 µm thickness on a vibratome (Leica VT-1200S). Tissue sections were placed on glass slides and coverslipped with Vectashield DAPI HardSet Mounting Medium (Vectorlabs). Images were collected at ~5× magnification using a fluorescent light stereo microscope (Olympus MVX10 or Nikon Eclipse Ti inverted widefield).

## Reach-to-grab task

As described previously (*Guo et al., 2015*; *Sauerbrei et al., 2020*), mice undergoing behavioral training were acclimated to head restraint and food-restricted to 80–90% of original body weight by limiting food intake to 2–3 g/day. Otherwise, mice had ad libitum food. Animals were monitored daily by veterinary staff, according to IACUC guidelines. During training, animals learned to reach for pellets of food with the right paw following an acoustic cue. In most sessions, the pellet was delivered with a rotating table (Janelia Experimental Technology), as described previously (*Guo et al., 2015*; *Sauerbrei et al., 2020*). In n = 7 sessions with DCN recording, food was instead delivered on an automated vertical post (Janelia Experimental Technology). Two high-speed cameras (Point Grey Flea3) with manual iris and focus lenses (Tokina 6–15 mm f/1.4, or Tamron 13VM1040ASIR 10–40 mm, f/1.4) were placed in front and to the right of the animal. A custom-made infrared LED light source was mounted behind each camera. Video was recorded at 500 Hz using BIAS acquisition software (IO Rodeo, *Will, 2020*). The cameras, acoustic cue and table, and laser were controlled using Wavesurfer software (Adam Taylor, Janelia Scientific Computing; http://wavesurfer.janelia.org/; *Taylor et al., 2020*) and a custom Arduino controller (Peter Polidoro, Janelia Experimental Technology).

## Video analysis

The position of the hand was tracked using the APT software package (https://github.com/kristinbranson/APT, *Lee et al., 2021*), developed by the Branson Lab at Janelia, as described previously (*Guo et al., 2015*; *Sauerbrei et al., 2020*). The position of the hand was manually annotated for training frames, a tracker was created using the cascaded pose regression (*Dollár et al., 2010*) or DeepLabCut algorithm (*Mathis et al., 2018*), and the tracker was applied to all movies in each dataset. The three-dimensional (3D) position of the hand was triangulated by performing a stereo calibration of the pair of cameras using the Caltech Camera Calibration Toolbox for Matlab (http://www.vision.caltech.edu/bouguetj/calib_doc/, *Bouguet, 2015*). The timing of behavioral waypoints, including lift and grab, was estimated with the Janelia Automatic Animal Behavior Annotator (https://github.com/kristinbranson/JAABA, *Kabra et al., 2021*), as described previously. Lift was defined as initial separation between paw and perch. Hand-open was defined as fingers beginning to separate from palm. Grab was defined as paw moving downward as digits closed. Supination was defined as wrist rotation >90° upward. At-mouth was defined as the frame in which the paw was within 1 pixel of mouth. Chew was defined as mastication of pellet visible in mouth. Each behavior classifier inputs a short sequence of frames from both the front- and side-view videos, and outputs a prediction of whether the mouse is performing the given behavior or not in the center frame (note that a different classifier is trained for each behavior and each mouse). The classifier used histogram of oriented gradient and histogram of optical flow features, general-purpose features that represent the directions and magnitudes of edge and motion vectors. Following the initial classification, post-processing was performed in which the per-frame classification results were smoothed by filling short gaps, and spurious short detection bouts were removed. We further filtered the detected events by considering

only event sequences in which a lift, hand-open, and grab were detected in order. Using the JAABA interface, the user manually checked for and corrected classifier errors and retrained the classifier, if necessary. Trials were regarded as 'single-reach successes' if the first lift-hand-open-grab sequence resulted in the animal grabbing the pellet, bringing it to the mouth, and chewing. Trials were regarded as 'multi-reach successes' if the animal missed on the first attempt, but subsequently grabbed the pellet, brought it to the mouth, and chewed.

## Electrophysiological recordings from the PN

On the day prior to recording, a craniotomy was made over the PN (A/P: −3.5 to 4.0 mm, M/L: 0.2–0.5 mm, D/V: 5.5–6.0 mm) in Sim1-Cre X Ai32 mice (n = 6) and sealed with silicone elastomer (Kwik-Sil, WPI). On the recording day, the animal was head-fixed, a fiber-coupled to a 473 nm laser (LuxX 473–80, Omicron Laserage) was placed over forelimb motor cortex, and a 384-channel Neuropixels 3A probe (https://www.neuropixels.org/) coated in a fluorescent dye (CM-DiI, DiI, DiO, Thermo Fisher Scientific; JF-669, Tocris) was slowly lowered into the brain. As the probe approached the PN, layer 5 motor cortical neurons were activated with a 10 Hz train of 5 ms pulses (2–10 mW), and the depth profile of evoked spiking was examined online. When it was determined that the bottom ~1 mm of the probe was in the PN, the craniotomy and recording probe were sealed with Kwik-Sil, and after 15 min the reaching task was initiated. Neuropixels data and timestamps for the camera and laser were acquired with a custom FPGA-based system. For offline assessment of the depth profile of evoked multiunit activity, the raw voltage data for each channel were blanked from 0 to 6 ms following pulse onset to remove the optical artifact, high-pass filtered with a cutoff of 650 Hz, full-wave rectified, averaged over all pulses, and smoothed over time with a σ = 333 μs Gaussian kernel. Next, the data were spatially smoothed across the probe using a σ = 30 μm Gaussian kernel, and the depth profile averaged between a temporal offset of 6–10 ms from pulse onset was plotted (*Figure 2B*, red curve). Once we verified, using the depth profile from the raw data, that the probe had been targeted to motor-cortex-recipient regions of the PN, we proceeded to analyze the activity of single units, which we sorted using JRClust (https://github.com/JaneliaSciComp/JRCLUST, *Jun et al., 2021*). Units on the bottom 1 mm of the probe were included for analysis; this is the region that the histology and depth profiles suggested to be in the PN. Firing rates were visualized by smoothing the cue-aligned spike trains using a σ = 50 ms Gaussian kernel, z-scoring based on the mean and standard deviation of the firing rate from cue −1000 ms to cue −200 ms, and averaging across trials (*Figure 2E*). For the tagging of PN neurons based on short-latency responses to the cortical pulses, we performed a signed-rank test on the paired spike counts in the 20 ms preceding and the 20 ms following the pulses (n = 30/129 neurons tagged, blue bar in *Figure 2E*). For the long laser stimuli (*Figure 2G*, upper, n = 80/129 neurons tested), spike counts were compared between the 2 s laser-on period and the 2 s pre-laser period using a two-sided signed-rank test. Multiple comparisons were corrected for by controlling the false discovery rate at 0.05 (*Benjamini and Hochberg, 1995*). Firing rates were classified as changing during the laser if the corrected p-value q < 0.05.

## GLM for cue and reach responses in the PN

In order to disentangle the effects of the cue and movement on PN activity, we fit a GLM to spiking activity for each PN neuron (*Park et al., 2014*). In this model, spikes are generated by an inhomogeneous Poisson process with intensity given by:

$$\lambda(t) = \exp((k_{cue} * x_{cue})(t) + (k_{reach} * x_{reach})(t)),$$

where $x_{cue}$ and $x_{reach}$ are delta functions centered at cue and lift times, respectively, $k_{cue}$ and $k_{reach}$ are linear filters describing the time-varying effects of each event type, and ★ denotes convolution. The filters $k_{cue}$ and $k_{reach}$ were constructed using half-cosine basis functions. Eight cosine bumps were evenly spaced across an 800 ms window starting at the event time. The model was fit by maximizing the log posterior with ridge regularization on the weights using the neuroGLM package for Matlab (https://github.com/pillowlab/neuroGLM; *Park et al., 2014*). We assessed the goodness-of-fit ($R^2$) by regressing the observed firing rates (Gaussian smoothing with σ = 50 ms) on the estimated firing rate λ(t). We visualized the estimated effects of the cue and reach on PN activity by plotting $\exp(k_{cue})$ and $\exp(k_{reach})$ for the neurons with $R^2 > 0.1$ (*Figure 2F*, three upper left panels and lower

two heatmaps). To assess the effects of cue and reach, we computed the Euclidean norm of $k_{cue}$ and $k_{reach}$ and compared their distributions (*Figure 2F*, upper right).

## Optogenetic stimulation of the PN

Selective expression of ChR2 was induced in the PN by injecting Slc17a7-Cre mice (n = 15) with AAV-2/1-CAG-flex-ChR2-TdTomato in the left PN and implanting a tapered optical fiber over the injection site, as described above. Animals were trained on the reach-to-grab task, and craniotomies over either forelimb motor cortex, the DCN (interpositus nucleus), or both were performed before the first day of recording. During the recording session, an optical coupler attached to a 473 nm laser (LuxX 473–80, Omicron Laserage) was attached to the connection on the top of the skull leading to the tapered fiber. During the experiment, three types of trials were administered: (a) control trials, in which a cue was given and the food pellet delivered, (b) laser-only trials, in which a 2 s, 10 Hz, 20 Hz, or 40 Hz sinusoidal laser stimulus was given with no cue, and (c) laser-cue trials, in which the sinusoidal laser stimulation was given either synchronously with the cue, 200 ms before the cue, or 300 ms before the cue. In most sessions (n = 38/45), 40 Hz stimulation was used, but in some (n = 7/45), 10 or 20 Hz stimulation was applied because pilot behavioral experiments on those mice suggested that lower-frequency stimulation might have larger effects on behavior. The laser power at the tip of the coupler was measured at 8–20 mW, but there was likely a significant power drop between the coupler and the fiber in the brain.

## Behavioral effects of PN stimulation

To directly visualize the effect of PN stimulation in the raw video frames, we compiled the video frames at offsets of 50, 150, and 250 ms from lift, and computed the 90th percentile of image intensity for each pixel on control trials and laser trials, resulting in one frame template for control and laser at each temporal offset. Then, we used the control and laser templates as pixelwise weights for orange (RGB = [1. 5 0]) and blue (RGB = [0. 5 1]), respectively, and added these weighted values together to produce the final images (*Figure 5A*). To compare control and laser conditions continuously over time, the same procedure was followed for every frame aligned to movement onset (*Video 1*) or laser onset (*Video 2*). Hand trajectories were plotted as a function of time (*Figure 5B*), and the position of the hand at the time of grab was extracted for control and laser plots and compared by plotting (*Figure 5C*). In order to summarize the effect of PN stimulation for each session, we used five kinematic and task variables, as follows. To correct for multiple comparisons across the 45 sessions, the Benjamini-Hochberg correction was applied to the p-values from the statistical tests for each measure, and we rejected the corresponding null hypotheses if the corrected p-value q < 0.05.

1. Difference in the median position of the hand at grab on laser and control trials (*Figure 5D*; two-sided rank sum test for each spatial direction, q < 0.05).
2. Median time from lift to grab on control and laser trials (*Figure 5E*; two-sided rank sum test, q < 0.05).
3. Probability of initiating a lift on control and laser trials (*Figure 5F*; chi-square test, q < 0.05).
4. Success rate on the first reach attempt on control and laser trials (*Figure 5G*; chi-square test, q < 0.05).
5. Standard deviation in the position of the hand at grab for laser and control trials (*Figure 5H* shows the sum of the standard deviation across the three spatial directions; for each direction, a two-sample, two-sided F-test for equal variances was performed, q < 0.05, and dots are blue for sessions in which the null hypothesis was rejected for at least one direction).

We were unable to make reliable comparisons of cue-to-reach reaction time on laser and control trials across sessions, due to three technical issues. First, in some sessions in which we delivered the pellet on a vertical post, the reaction times were negative on control trials, because the animal could hear the post raising up and reached before the acoustic cue was given. Second, we imposed a delay from the laser onset to the cue in several sessions, because we had previously observed an apparent tendency for animals to reach at the laser onset, and wanted to verify that animals were not using the laser as a cue. Third, PN stimulation blocked movement initiation in some sessions, which resulted in an uninterpretable reaction time on laser trials for those sessions. Because these

issues complicated the interpretation of cue-to-reach reaction time, we did not include it as a variable of interest.

## Electrophysiological recordings in the cerebellum and motor cortex

At the start of the optogenetic stimulation experiments (n = 45 sessions, n = 15 mice), four-shank, 64-channel probes (Janelia Experimental Technology) were lowered to approximately 0.9 mm below the surface of forelimb motor cortex, 2.5 mm below the cerebellar surface (for DCN recording), or both. In total, we obtained recordings from 1157 cortical neurons across 38 sessions in 10 mice and recordings from 139 DCN neurons across 12 sessions in five mice. For five of these sessions in three mice, both areas were recorded simultaneously. For the DCN (interpositus nucleus), the probe was lowered until large, high-frequency spikes were visible and complex spikes (characteristic of Purkinje cells in the cerebellar cortex) were no longer present. In some cases, the probe was lowered past the DCN and was then retracted until spiking reappeared. Purkinje cells (n = 23 neurons in n = 2 mice) were recorded in Crus I, Crus II, and the paramedian lobule. The 16 recording sites on each shank had a range of 0–320 μm from the tip of the shank. Spike sorting was performed with JRClust or Kilosort2 (*Stringer et al., 2019*; https://github.com/MouseLand/Kilosort2, *Pachitariu et al., 2020*).

In order to assess whether cortical neurons were modulated during movement and by PN stimulation, we compared paired spike counts for each trial immediately before and after lift or laser onset (using laser-only trials) with a two-sided signed-rank test. The pre- and post-lift windows were (−1000, –500) and (−50, 450), and the pre- and post-laser windows were (−150, 0) and (0, 150) ms. A Benjamini-Hochberg correction for multiple comparisons was applied, and neurons were classified as modulated by the lift or laser if the corrected p-value q < 0.05. Event-aligned firing rates were visualized by smoothing the spike trains with a σ = 50 ms Gaussian kernel, z-scoring based on the mean and standard deviation of the firing rate within a window of (−1000, –400) ms of lift and (−1000, 0) of laser, and averaging across trials. For DCN neurons, the same procedure was used to characterize movement-aligned responses. Because the laser-only responses of DCN neurons were extremely variable, exhibiting a mixture of transient and tonic increases and decreases, we could not classify them into tagged and untagged groups, as we did for cortical neurons. For both cortex and DCN neurons, we computed the Spearman correlation in firing rate z-scores on control and laser trials at each time point across all neurons (*Figures 6C* and *7C*, upper insets), and the Spearman correlation across all time points for each individual neuron (*Figures 6C* and *7C*, right insets). Note that these correlations were computed on z-scores, rather than the raw firing rates; had we taken the latter approach, we would have observed positive correlations before movement onset in *Figures 6C* and *7C*, upper, because baseline firing rates are correlated in control and laser conditions. In order to determine whether the firing rate changes following laser-only stimulation were related to firing rate changes induced by laser stimulation during the movement, we predicted what effect the laser would have had on neural activity if it induced the same response during reaching as during laser-only stimulation. To do that, we took the peri-laser-only firing rate for each neuron, z-scored it using the pre-lift mean and standard deviation, offset it at the observed lift-to-laser delays for each trial in the corresponding session, and averaged across trials (*Figure 6—figure supplement 1E*, *Figure 7—figure supplement 1E*, lower heatmaps). We then computed the Spearman correlation between these predictions and the actual laser – control firing rate z-score differences across all neurons at each time point (*Figure 6—figure supplement 1D*, *Figure 7—figure supplement 1D*). The correlation values over time, along with the q-values for the test against the null hypothesis of zero correlation, are plotted in *Figure 6—figure supplement 1E* and *Figure 7—figure supplement 1E* (upper insets). Spike sorting for Purkinje cells was performed with custom Matlab scripts. Purkinje cells were identified based on depth, firing rate, and, typically, the presence of a complex spike.

## PN stimulation-evoked multiunit activity in the cerebellum

In order to determine whether the effects of optogenetic stimulation of the PN propagated into cerebellar cortex, we stimulated ChR2-expressing PN neurons with an implanted optical fiber and inserted a 384-channel Neuropixels probe into cerebellar cortex (bregma −7.0 mm, right 2.5 mm, depth 2.5 mm, with the probe 4° from vertical, so that the tip of the probe was anterior to the point at which it entered the brain). We then delivered light in a 40 Hz sinusoidal pattern to the PN (mean

power 3 mW). Because we were able to obtain very few stable, well-isolated neurons in the cerebellar cortex, we instead examined the spatio-temporal profile of multiunit activity, which likely reflects the combination of signals from mossy fiber terminals (including those from stimulated PN neurons), granule cells, Purkinje cells, and interneurons. Data from each site on the Neuropixels probe were high-pass filtered with a cutoff of 650 Hz, full-wave rectified, and smoothed over time with a Gaussian kernel ($\sigma$ = 333 $\mu$s). The resulting traces were spatially smoothed across the probe (Gaussian kernel, $\sigma$ = 30 $\mu$m), and the resulting spatio-temporal profile was visualized as a heatmap, with the channel-averaged signal and power density at 40 Hz for each site displayed as insets (*Figure 3—figure supplement 1B*).

## Latency analysis

To assess the response latency of PN neurons (n = 129) to pulse-train optogenetic stimulation of motor cortex, we computed the spike counts in a sliding 10 ms window with 1 ms increments from −30 to 50 ms of pulse onset for each PN neuron. At each lag, we compared the counts within each window with the counts in the pre-pulse bins using a rank sum test, then applied a Benjamini-Hochberg correction for multiple comparisons across neurons. Next, at each lag, we computed the fraction of neurons that had been recruited at or before that lag (*Figure 2—figure supplement 1A*). To better assess the latency distribution among responsive neurons, we also divided this recruitment curve by the fraction of responsive neurons from the entire sample (n = 51/129), resulting in a curve ranging from 0 to 1 (*Figure 2—figure supplement 1B*). (Note that this analysis to determine the timing of recruitment differs from the approach used to identify which neurons are tagged [blue inset in *Figure 2E*], which used paired spike counts in a more restricted window of 20 ms before and after the pulse.) For the tagging of motor cortical neurons, Purkinje cells, and DCN neurons from PN stimulation, we performed the same analysis aligned to the onset of the sinusoidal laser stimulus, using a sliding window of 10 ms in width in increments of 1 ms within 100 ms preceding and 100 ms following the stimulus.

## Neural entrainment to PN stimulation

The goal of this analysis was to determine whether spiking activity in Purkinje cells and DCN neurons was entrained to 40 Hz sinusoidal optogenetic stimulation of the PN. A standard approach to determining the synchronization of a point pattern (e.g., spike times) with a periodic time series (local field potential oscillations, locomotion, or – as in this case – rhythmic stimulation) is to regard the event times as circular data and to test whether the resulting phase distribution deviates from uniformity (*Fisher, 1995*). For each spike, we computed the phase of the sinusoidal stimulation at the corresponding time. To exclude distortion of the phase distribution from transient firing rate changes at stimulus onset, we used spikes occurring from 500 ms of stimulus onset to the termination of the stimulus at 2000 ms. For each neuron, we computed the phase distribution of spikes using a kernel density estimator with a bandwidth of $\sigma$ = 0.3 (*Muir, 2021*, code available at https://www.mathworks.com/matlabcentral/fileexchange/44072-kernel-density-estimation-for-circular-functions). We determined whether each neuron was entrained to the stimulation using a Rayleigh test (*Berens, 2019*, code available at https://github.com/circstat/circstat-matlab), with a Benjamini-Hochberg correction for multiple comparisons (rejecting the null hypothesis of uniformity for q < 0.05) and computed the mean resultant to determine the phase preference.

## Neural decoding

We designed linear filters to decode 3D hand velocities from neural activity recorded in motor cortex and DCN, to assess whether the observed behavioral differences between control trials and trials with PN perturbation ('laser trials') were correlated to corresponding differences in neural activity. We excluded trials in which the animal did not initiate a reach, and considered a window of time from 100 ms before lift to 300 ms after the lift for the other trials. To guarantee that decoding results for laser trials reflect movements performed with PN perturbed, we excluded laser trials in which the lift-to-grab sequence was not initiated and completed while the laser was turned on or in which the lift occurred less than 300 ms before the end of the laser period. Two sessions with DCN recordings had less than two laser trials satisfying these criteria, so they were not included in the decoding analysis. For sessions in which the pellet was delivered on the rotating table, we also excluded atypical

control trials in which the lift-to-grab sequence was not completely included within the 2 s following the acoustic cue. For sessions in which the pellet was delivered on a vertical post, we used a similar criterion but also allowed the lifts to occur up to 0.5 s before the cue because the motor moving the pellet started earlier than the acoustic cue, and some animals effectively cued their movement to the motor. Following our prior work (*Sauerbrei et al., 2020*), the decoders use multiunit neural activity (either threshold crossings for each channel, when JRClust was used, or manually curated multiunits, when Kilosort2 was used), preprocessed as follows: (a) counts of detected spikes are smoothed with a Gaussian kernel with $\sigma = 25$ ms, (b) these smoothed firing rates are z-scored with respect to the activity at rest (1.5 s window preceding the start of each trial) and channels with mean absolute z-scores greater than 100 during movement are excluded, (c) these z-scores are processed with PCA, with the principal components computed from the lift-aligned trial-averaged activity in control trials. At any instant of time, the decoders use the 15 most recent samples (hence up to 28 ms in the past) of PCA-reduced neural activity in cortex or DCN, to decode the hand velocity at that time. The following procedure was used to choose the number of PCA dimensions used by the decoder and the decoder coefficients in each experimental session. First, one-fifth of the control trials ('control test' set) was kept aside for testing the performance of the decoder, thus enabling fair comparison with decoder performance in laser trials. On the remaining set of control trials, fourfold cross-validation was performed with an increasing number of PCA dimensions. For each fold and each number of PCA dimensions, decoder coefficients were computed by regressing observed hand velocity data against PCA-reduced neural data in three-fourths of the trials, and then used to predict the velocities in the left-out trials. Hand velocities were computed by smoothing the raw 3D hand trajectories with a Gaussian kernel ($\sigma = 25$ ms, as for neural activity) and then applying a central difference filter of order 8. For each number of PCA dimensions, we compared the average across folds of the mean squared error between predicted and observed velocities. We finally chose the minimum number of PCA dimensions that guaranteed performance within 1% of the overall minimum across all choices, and averaged the decoder coefficients for that number of PCA dimensions across the folds. The decoder thus derived was applied to the neural activity data in the control test trials and the laser trials to predict the hand velocities with and without PN perturbation and contrast these predictions with the hand velocities observed experimentally (*Figure 9A*, *Figure 9—figure supplement 1A–B*).

We applied the decoding analysis to all experimental sessions with cortical recordings (n = 38) and with DCN recordings (n = 10) independently, choosing in each case the region of the dataset that maximized the number of stable channels (from cortex or DCN). In the last few recording sessions (n=5, indicated by gray circles in *Figure 9—figure supplement 1C–J*), all using the table for pellet delivery, cortical and DCN recordings were collected within the same experiment. Incidentally, the yield of cortical multiunits in these experiments was among the best across all sessions (likely due to the use of Kilosort2 for spike sorting) so the baseline performance of the cortical decoder in these sessions was better than in most other sessions. To summarize the decoding performance in each session, we used the coefficient of determination ($R^2$) between decoded and observed hand velocities, after pooling all directions together. The velocity decoding accuracy was higher in control test trials than in laser trials in most of the cortex decoding sessions, and all of the DCN decoding sessions (*Figure 9—figure supplement 1C–D*). This drop in $R^2$ on laser trials could be due to actual differences in how the neural signals are transformed into behavior (e.g., via other compensatory pathways active when PN are perturbed) but could also partly reflect generalization error. In fact, the decoder was trained only on control trials, and therefore a drop in decoding performance was expected for a trial type not used for training. We investigated this possibility by repeating the decoding analysis with an alternative decoder trained on a perfectly balanced set of control and laser trials. This 'balanced decoder' was similar to the original decoder, but: (a) the training set consisted of all odd laser trials and a matching number of control trials, sampled at regular intervals of time from within the set of all control trials, (b) PCA components were computed from the average of the lift-aligned trial-averaged activity in laser trials and the lift-aligned trial-averaged activity in control trials, (c) the number of PCA dimensions used by the decoder was fixed to five rather than cross-validated (due to the limited number of available trials after setting aside laser trials for the test set, and subsampling control trials to match the number of laser trials), (d) the decoder performance was tested on the laser and control trials not used for training the decoder. Note that while in the original decoder the test set is roughly balanced between laser and control trials, for the balanced decoder, it is the training set that is exactly balanced, but the test set contains many more

control trials than laser trials. When we compared the balanced decoder performance on control and laser test trials, we did not find statistically significant differences in either cortical or DCN-based decoding (*Figure 9—figure supplement 1*), suggesting that in both regions there is a neural population that can explain the behavior during PN perturbation as well as in control trials, and the difference in performance in the original decoder may in fact be due to generalization error.

We performed additional analyses with the initial decoder design (trained only on control trials) on the simultaneous cortical and DCN recordings (n = 5 sessions from n=3 animals). First, we repeated the computation of the decoder coefficients for the cortex-based and the DCN-based velocity decoders (each using the cross-validated number of PCA dimensions found in the original decoding analysis) after excluding trials that were not available in both datasets, to guarantee that both decoders used the same training set. Similarly, we used an identical set of control test and laser trials for all decoders. In addition to comparing the cortex-based and DCN-based decoding performance (e.g., *Figure 9A* for one of the sessions), we compared the decoding performance from each individual region with that of a third decoder that used both cortical and DCN activity. This 'augmented decoder' (CTX+DCN) decoded 3D hand velocities from the PCA dimensions of activity in cortex and DCN concatenated with each other. In all the sessions with simultaneous recordings, the decoding performance of the cortex-based decoder was better than that of the DCN-based decoder, but this result was confounded by the superior recording yield in cortex compared to DCN. More interestingly, the performance of the 'augmented decoder' (CTX+DCN) was almost identical to that of the cortex-based decoder in control trials (*Figure 9—figure supplement 1E*), and slightly worse in laser trials (*Figure 9—figure supplement 1F*). A similar analysis of the sessions with simultaneous cortical and DCN recordings was also repeated using the 'balanced' version of the decoder – with five PCA dimensions from each region – and even in that case the augmented decoder did not outperform the cortex-based one (*Figure 9—figure supplement 1I–J*). This lack of decoding performance improvement over the cortical decoder suggests that DCN contains behaviorally relevant dimensions of activity similar to cortex, at least within the constraints of our single-target reaching task.

## CCA for motor cortex and DCN

For the simultaneous cortical and DCN recordings (n=5 sessions), we also performed CCA to assess the similarity in population activity between the two regions during the movement on a single-trial level. Following the cross-validated PCA reduction of neural data performed within the initial decoding analysis, four (or more) PCA dimensions of activity were available in both cortex and DCN for each of the sessions with simultaneous recordings. Therefore, we used the first four dimensions of activity within each region. For each dimension, we concatenated the PCA scores on all control and laser trials in the window [lift−100 ms, lift+300 ms], and centered the resulting vector. We then performed CCA between the matrices of concatenated activity in cortex and DCN. This enabled us to obtain linear combinations of the original principal components in each region ('canonical variates') whose concatenated trajectories ('canonical scores') were maximally correlated between the two regions. The canonical scores are shown for each trial separately in *Figure 8A*. The first pair of canonical variates ('Cortex dim.1' and 'DCN dim.1' in the figure) are those maximally correlated between the two regions. The second pair of canonical variates ('Cortex dim.2' and 'DCN dim.2') are those maximally correlated with each other while being uncorrelated to the first pair, and so on for the other two pairs of variates. *Figure 8B* shows the correlation values for each pair of canonical variates, revealing that in each session CCA found two dimensions of well-correlated activity between cortex and DCN (correlations 0.58–0.88), whereas the remaining dimensions were weakly correlated (correlations<0.27). Differently from PCA, in which components are ordered based on the fraction of variance they explain in the original data, the ordering of canonical variates is purely based on correlations and agnostic to variance explained. Therefore, we next assessed whether the first two canonical variates found by CCA in each region, those well correlated to corresponding canonical variates in the other region, explained a large or small fraction of the total variance within their own region. For that, we computed the proportion of (standardized) sample variance explained by each canonical variate, as in *Johnson and Wichern, 2007*, and plotted the cumulative fraction of variance explained by an increasing number of canonical variates (*Figure 8C*, upper left and bottom left panels). In cortex, the first two canonical variates explained between 49% and 64% of the variance (depending on the session), whereas in DCN the first two canonical variates explained between

51% and 70% of the variance. Hence, while the dimensions of cortical and cerebellar activity that are strongly correlated with each other usually explain the majority of the variance, the fraction explained by dimensions of activity that are weakly correlated across regions is 30% or more, and thus not negligible. For each canonical variate, after computing the fraction of variance it explains within its own region, it is straightforward to compute the fraction of variance it explains within the opposite region. Since each canonical variate is only correlated to one of the canonical variates of the other region, this is simply the fraction of variance it explains within its own region times the square of the canonical correlation with the corresponding variate in the other region. We thus computed these quantities for each canonical variate and plotted the cumulative fraction of variance explained in each region by an increasing number of canonical variates of the other region (*Figure 8C*, upper right and lower right panels). When the complete set of canonical variates is considered (#canonical variates=4 in the figure), we obtain the total proportion of (standardized) sample variance in one region explained by the other region, a compact metric for the similarity (or redundancy) in activity. In cortex 26–47% of variance was explained by the canonical variates in DCN (depending on the session), whereas in DCN 25–50% of variance was explained by the canonical variates in cortex.

Finally, we investigated whether the similarity in activity between cortex and DCN is affected by the PN perturbation. For this analysis, we could not rely on the CCA discussed above because control trials were much more numerous than laser trials, and PCA components themselves were computed based on trial-averaged activity in control trials only (as per the initial decoder design). Instead, we performed two additional CCAs, both using PCA components computed on trial-averaged activity in control and laser trials (as per the balanced version of the decoder). The first of these CCAs used only the concatenated activity (first four PCA components) recorded in laser trials, whereas the second one used only the activity recorded in a number of control trials matching the number of laser trials (sampled at regular intervals of time from within the set of all control trials). In each session, we then compared the similarity in population activity between cortex and DCN, quantified by the total proportion of variance in one region explained by the other region, for the CCA using only laser trials and the CCA using only control trials. The scatter plots in *Figure 8D* (left=cortex, right=DCN) summarize the outcome of this comparison across sessions. In four out of five sessions, our similarity metric was lower for the CCA on laser trials than the CCA on control trials, but the average difference across sessions was small (5.8% reduction of total variance in cortex explained by DCN, 4.4% reduction of total variance in DCN explained by cortex, for laser vs. control). Overall, the effect of the PN perturbation on the similarity in activity between cortex and DCN appears to be mild.

## Acknowledgements

We thank Gülşen Sürmeli, Diana Burk, and Cheng-Chiu Huang for help with pilot experiments, Adam Taylor for Wavesurfer software, Peter Polidoro for instrumentation development, Tim Harris and the Neuropixels Consortium for probe development, James Jun and Marius Pachitariu for spike sorting software, Sal DiLisio and the Janelia Vivarium for viral injections and implantation surgeries, the Janelia Histology Core for histology and imaging support, the Janelia Visitor Program for supporting pilot experiments, and Yifat Prut, Abdulraheem Nashef, Steve Edgley, Amy Bastian, Stephen Scott, Daniel Wolpert, and Brett Mensh for discussions.

## Additional information

### Funding

| Funder | Author |
| --- | --- |
| Howard Hughes Medical Institute | Adam W Hantman |

The funders had no role in study design, data collection and interpretation, or the decision to submit the work for publication.

## Author contributions
Jian-Zhong Guo, Conceptualization, Data curation, Investigation, Methodology, JG performed behavioral experiments, cerebellar recordings, and cortical recordings and analyzed behavior data; Britton A Sauerbrei, Conceptualization, Data curation, Formal analysis, Supervision, Visualization, Writing - original draft, Writing - review and editing, BS analyzed electrophysiology and behavior data, interpreted the results, and wrote the manuscript; Jeremy D Cohen, Data curation, Investigation, Methodology, Writing - review and editing, JC performed Neuropixels recordings in the PN and cerebellar cortex; Matteo Mischiati, Conceptualization, Formal analysis, Visualization, Writing - original draft, Writing - review and editing, MM developed and performed the neural decoding and canonical correlation analyses; Austin R Graves, Investigation, AG performed preliminary slice electrophysiology experiments and analyzed preliminary behavioral data; Ferruccio Pisanello, Resources, Methodology, FP developed and fabricated the optical fibers for PN stimulation; Kristin M Branson, Data curation, Software, Formal analysis, Writing - original draft, KB developed video analysis tools and analyzed preliminary behavior data; Adam W Hantman, Conceptualization, Supervision, Funding acquisition, Writing - original draft, Project administration, Writing - review and editing, A.H. supervised the project, interpreted the results, and wrote the manuscript

## Author ORCIDs
Jian-Zhong Guo (iD) https://orcid.org/0000-0002-5836-5921
Britton A Sauerbrei (iD) https://orcid.org/0000-0003-3386-3243
Jeremy D Cohen (iD) https://orcid.org/0000-0003-4961-222X
Matteo Mischiati (iD) https://orcid.org/0000-0002-8448-9143
Ferruccio Pisanello (iD) https://orcid.org/0000-0002-1489-7758
Adam W Hantman (iD) https://orcid.org/0000-0002-6563-1423

## Ethics
Animal experimentation: All procedures were performed in accordance with protocols approved by the Institutional Animal Care and Use Committee (IACUC) of the Janelia Research Campus (protocols 16-139 and 19-177).

## Decision letter and Author response
Decision letter https://doi.org/10.7554/eLife.65906.sa1
Author response https://doi.org/10.7554/eLife.65906.sa2

# Additional files

## Supplementary files
• Supplementary file 1. Table of behavioral effects of optogenetic pontine nuclei (PN) stimulation for each experimental session.
• Transparent reporting form

## Data availability
Data and code are available at Dryad (https://doi.org/10.5061/dryad.mgqnk990f).

The following dataset was generated:

| Author(s) | Year | Dataset title | Dataset URL | Database and Identifier |
| --- | --- | --- | --- | --- |
| Sauerbrei BA, Guo J, Cohen J, Mischiati M, Graves A, Pisanello F, Branson K, Hantman A | 2021 | Data supplement for "Disrupting cortico-cerebellar communication impairs dexterity" (Guo*, Sauerbrei* et al., eLife 2021) | https://doi.org/10.5061/dryad.mgqnk990f | Dryad Digital Repository, 10.5061/dryad.mgqnk990f |

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
