## [Decision Letter]

**Acceptance summary:**

This paper uses high density physiological recordings from multiple brain areas associated with reaching movements to assess how large brain networks coordinate action. The results contribute to a long line of work supporting the view that the cortico-cerebellar pathway is required for fine motor control. Moreover, optogenetic perturbation of precerebellar neurons in the pontine nuclei demonstrates that perturbation of a single source of mossy fiber input is sufficient to perturb dexterous skilled forelimb movements in mice.

**Decision letter after peer review:**

Thank you for submitting your article "Dynamics of the Cortico-Cerebellar Loop are Necessary for Fine-Tuning Dexterous Movement" for consideration by *eLife*. Your article has been reviewed by 3 peer reviewers, and the evaluation has been overseen by a Reviewing Editor and Tirin Moore as the Senior Editor. The reviewers have opted to remain anonymous.

Essential Revisions:

The Reviewers agreed that the data were comprehensive and of interest, as it shows that PN perturbation disrupts activity across a cortical cerebellar loop in ways that are commensurate with observed degradation in movement accuracy. However, there was agreement that the current presentation often goes beyond what the data actually support; major conclusions in the abstract, intro, and results are not directly linked the results that are actually presented. Tempering down some of the claims and substantive additional analyses and revisions to the text/figures are required. In particular, while the paper suggests a distinction between activity driving motor execution vs. fine tuning movement accuracy, the reviewers felt that the current data has limited power in disentangling these processes and does not yet clarify the precision function of the cortico-cerebellar loop in controlling movement accuracy. Finally, there was agreement that the findings need to be better contextualized within the existing literature in order to more appropriately motivate the authors' specific experimental design and clarify the actual conceptual advance.

*Reviewer #1 (Recommendations for the authors):*

1. In figure 8 GLM DCN activity/kinematics correlations appear bimodal for the outward direction. Was this distribution accounted for in the statistical treatment, where difference from the zero is measured in one direction? This point might be interesting in light of causal roles of DCN activity in adjusting limb kinematics.

2. Histology of recording sites in the cerebellum and cortex should be shown, and ideally related to the activity recorded there.

3. The authors should address whether there are any motor cortical neurons that respond to just the cue, to support their conclusion that PN neurons integrate signals of multiple modalities. It is difficult to relate data in Figure 2 showing PN recordings, to data in Figure 4 showing M1 recordings, since alignment is different, to cue vs to lift. If a subset of M1 neurons respond to cue selectively, might the PN neurons be integrating cue and motor related M1 responses? I don't doubt the conclusion that there is integration in the pons, but would be valuable to better understand any principles of this integration, ie whether it is derived from sensory plus motor areas, or sensory-responsive motor areas. Short of more analyses, even more discussion of what the authors think about the significance of this integration would be useful.

[Editors' note: further revisions were suggested prior to acceptance, as described below.]

Thank you for resubmitting your work entitled "Disrupting cortico-cerebellar communication impairs dexterity" for further consideration by *eLife*. Your revised article has been reviewed by 3 peer reviewers and the evaluation has been overseen by a Reviewing Editor and Tirin Moore as the Senior Editor.

The manuscript has been improved but there are some remaining issues that need to be addressed, as outlined below:

Please address the remaining concerns of Reviewers 1 and 3.

*Reviewer #1 (Recommendations for the authors):*

The authors have made a strong effort in improving this study. While the study remains a bit unusual in its focus, I think the data are important and will be of interest. I recommend the following editorial changes:

1. Line 141. Add a potential interpretation of the cue responsive pons neurons as potentially reflecting preparatory activity, since the cue predicts the movement.

2. Figure 3 rasters are really pointless as shown. They look like solid gray bars. Historically several tricks have been used to show these high firing rate cells on per-trial basis. One is to show every x spike (where x is 3rd or 4th) and report that. This sparsens the signal a bit so that you can appreciate the modulation. The second is to compute instantaneous rates and illustrate in a heat map for each trial. Both of these methods would improve exposition of the point the authors are, I think, trying to make: that the response is reliable across trials. Right now the figure is ineffective at showing this, despite the changes purportedly made in response to the first round of reviews on this point.

3. The DCN lag expressed in degrees should include time, since the cycle is 25 ms I presume (40 Hz). E.g. DCN entrainment lagged Purkinje cells by __ ms (95 degrees).

4. The CCA analysis is novel and interesting and will be of importance to neuroscientists broadly. This seems like the most significant step of the study. It should be moved to a main figure rather than supplemental figure 7.3. (If fewer main figures are needed, Figure 1 could be a supplement.)

5. Ln 831 "Decoding of hand velocity from neural activity on control and PN stimulation trials": I appreciate the authors now asking whether kinematic decoding can improve by including DCN with M1 activity. The new data are said to show that this does not improve decoding over cortex alone. It would be helpful to follow the logic if the authors explained/accounted for the large difference in R-squared between the first analyses using the DCN or M1 alone with values at 0.55 vs those in the pooled case where the R2 is 0.78. It is difficult to appreciate the negative result with this baseline shift being so large.

6. Discussion Paragraph line 448 – it should be mentioned that endpoints were hypermetric or hypometric with interposed activation and inactivation in Becker and Person 2019 and hypermetric with interposed ablation (Low.… Chen et al., 2018), not just that the velocities changed. Additionally, work in mice corroborated earlier studies in cats which showed endpoint problems with muscimol inactivation of the interposed nucleus (Cooper, Martin, Ghez 2000) which should be cited.

7. Lines 450-452: The wording seems off: Rather than "taken together with these previous studies, our findings suggest" I would strongly recommend stating: "Consistent with these previous studies our findings support the idea that the cortico-cerebellar loop…" This recommendation is premised on the connotation of the original text that fine-tuning was not a conclusion previously drawn about cerebellar contribution to movement; to the contrary, it is a major and consistent theme, repeatedly brought up. The present data support those notions and are consistent with them.

*Reviewer #2 (Recommendations for the authors):*

The revised manuscript has addressed all my comments. The paper is substantially improved. The manuscript now more explicitly discusses distinctions between driving vs. fine-tuning movement in the context of previous motor cortex and cerebellum manipulations that together clarify the rationale for the current study. The text now more closely mirrors the data, and potential limitations are more clearly discussed. The revisions to data presentation now provide a better view of the data. The latency analysis and canonical correlation analysis are great additions that strengthen the overall conclusion. The study provides compelling data dissecting the contribution of a cortico-cerebellar loop to fine motor control. This paper will be a valuable addition to the literature.

*Reviewer #3 (Recommendations for the authors):*

The authors have made major revisions to the manuscript and have been largely responsive to reviewer comments. They now more clearly motivate the novelty of the present study in the context of the existing literature and have revised the Results to better align their concluding sentences with the actual results that are shown. The addition of histograms to Figures 5D-h are helpful in ore directly describing the effects of stimulation.

To address reviewer concerns, the authors have also completed new analyses the latency of neural responses (Purkinje cells, DCN neurons, and motor cortical neurons) to optogenetic PN. I had assumed that quantification of latencies of the effects that are observes in the different nodes of the cortico-cerebellar loops would strengthen the authors' conclusion that they are actually studying the direct loop in Figure 1 which would then make the study's conclusions more compelling.

As they detail in the results, the authors observe a logical increase in the latency of the activation along this pathway, for the small % of neurons (5%) with the shorted response times: 10ms for Purkinje cells, 13ms for DCN neurons, and 21 ms for cortical cells. They then acknowledge in their comments to the reviewers that the values – even for these fastest responding neurons – are longer than expected for pulse train stimulation. However, this point is not explained in the text.

Additionally (and more importantly) the authors do not address the clear question arising from their new analysis, namely why do the vast majority of cells have such long response times to activation? For example, the new quantification of latencies added to the revision shows that the response latencies of most Purkinje cells are >63ms. Do the authors believe that such responses are mediated via a direct projections from the PN? If yes, more explanation is needed. If not, what are the alterative possible explanation(s)?. Either way, this point should be addressed in the revised Discussion.

Finally, as noted in my original review, the present these results are similar to those previously reported in primate DCN cooling experiments characterizing changes in hand movement in in a voluntary tracking task (e.g., Brooks et al., 1973; Conrad and Brooks 1974). The impact of the manuscript would be improved by returning to these studies in the discussion, and explaining why/how the application of the current methods provides a conceptual advance.

---

## [Author Response]

Reviewer #1 (Recommendations for the authors):1. In figure 8 GLM DCN activity/kinematics correlations appear bimodal for the outward direction. Was this distribution accounted for in the statistical treatment, where difference from the zero is measured in one direction? This point might be interesting in light of causal roles of DCN activity in adjusting limb kinematics.

To determine whether the difference between decoded velocity on laser and control trials was correlated with the corresponding laser-control difference in the observed velocity, we used a two-sided sign rank test on the correlations for all datasets. This nonparametric test is based on ranks, and is therefore invariant to monotonically-increasing transformations of the data. Thus, bimodality will not influence the test statistic or p-value (in the sense that the data could be transformed to have a unimodal distribution without changing these). A direct assessment of the bimodality of the correlations for the DCN datasets in Figure 8C (e.g., using Hartigan’s dip test) is not feasible, due to the small sample size (n = 10 datasets).

2. Histology of recording sites in the cerebellum and cortex should be shown, and ideally related to the activity recorded there.

We now show the cerebellar histology in Figure 3—figure supplement 4. Because we made many electrode penetrations in each mouse for cortical recordings (which makes dye tracks from individual sessions difficult to see), and because forelimb motor cortex lies at the surface of the brain and can be targeted reliably using anatomical landmarks, we did not collect the histological sections at the cortical sites. Note that the red DiI often saturated the red / orange channel, and ChR-tdTomato+ mossy fibers are not visible with the imaging settings we used for several animals.

3. The authors should address whether there are any motor cortical neurons that respond to just the cue, to support their conclusion that PN neurons integrate signals of multiple modalities. It is difficult to relate data in Figure 2 showing PN recordings, to data in Figure 4 showing M1 recordings, since alignment is different, to cue vs to lift. If a subset of M1 neurons respond to cue selectively, might the PN neurons be integrating cue and motor related M1 responses? I don't doubt the conclusion that there is integration in the pons, but would be valuable to better understand any principles of this integration, ie whether it is derived from sensory plus motor areas, or sensory-responsive motor areas. Short of more analyses, even more discussion of what the authors think about the significance of this integration would be useful.

Ideally, we would approach this question by fitting a GLM for each motor cortical neuron, as for the PN data in Figure 2F. The GLM works reasonably well for PN neurons because the dominant signal in the pontine population is a very large firing rate increase at the cue. We tried this method for cortical neurons, however, and found that the model fits were extremely poor: only 50% of cortical neurons had R^2^ values over 0.1, in comparison with 80% of PN neurons. Because of this poor fit, the resulting filters are not very meaningful. We have tried to disentangle cue- vs reach-related activity in single motor cortical neurons, but have found this challenging, because many animals initiate the reach rapidly after the cue and with relatively low reaction time variance. At the population level, we have found that cortical activity changes relative to baseline are highest shortly after movement onset (Figure 7B in the current manuscript; Sauerbrei et al. 2020, Figure 1F), single-trial neural trajectories are aligned to movement onset (Sauerbrei et al., bioRxiv 2018, Figure 1f), and a rapid decrease in the variability of spike timing occurs at movement onset (Sauerbrei et al., bioRxiv 2018, Figure S1A). To rigorously disassociate cue from reaching responses for individual motor cortical neurons, it would be necessary to impose a relatively long and variable delay period (e.g., distributed uniformly on [1, 2] s).

How, then, should the PN responses to the cue be interpreted? Although many of our PN neurons were tagged by M1 stimulation, it is not clear that the cue-aligned responses are inherited entirely from M1. Instead, we believe these responses likely reflect a large convergence of input from many cortical regions. Previous work in anesthetized rats has demonstrated that individual PN neurons can respond to stimulation of such disparate cortical areas as somatosensory and visual cortex (Potter, Ruegg, and Wiesendanger, 1978). If, as seems likely, the acoustic cue induces synchronized auditory, attentional, and motor planning signals across many cortical regions, this might explain why so many PN neurons in our sample exhibit firing rate changes locked to the cue. As our experiments do not enable us to independently control these modalities (acoustic, attentional, and planning), we remain agnostic as to which modalities are reflected in the cue-locked PN responses.

[Editors' note: further revisions were suggested prior to acceptance, as described below.]

The manuscript has been improved but there are some remaining issues that need to be addressed, as outlined below:Please address the remaining concerns of Reviewers 1 and 3.Reviewer #1 (Recommendations for the authors):The authors have made a strong effort in improving this study. While the study remains a bit unusual in its focus, I think the data are important and will be of interest. I recommend the following editorial changes:1. Line 141. Add a potential interpretation of the cue responsive pons neurons as potentially reflecting preparatory activity, since the cue predicts the movement.

We have added the following sentence to the Results: “These cue-aligned changes in firing rate might reflect acoustic responses, changes in attention, motor planning, or some combination of these factors.”

2. Figure 3 rasters are really pointless as shown. They look like solid gray bars. Historically several tricks have been used to show these high firing rate cells on per-trial basis. One is to show every x spike (where x is 3rd or 4th) and report that. This sparsens the signal a bit so that you can appreciate the modulation. The second is to compute instantaneous rates and illustrate in a heat map for each trial. Both of these methods would improve exposition of the point the authors are, I think, trying to make: that the response is reliable across trials. Right now the figure is ineffective at showing this, despite the changes purportedly made in response to the first round of reviews on this point.

We now plot every third spike in these panels.

3. The DCN lag expressed in degrees should include time, since the cycle is 25 ms I presume (40 Hz). E.g. DCN entrainment lagged Purkinje cells by __ ms (95 degrees).

We now note in the text that this lag corresponds to 6.6 ms.

4. The CCA analysis is novel and interesting and will be of importance to neuroscientists broadly. This seems like the most significant step of the study. It should be moved to a main figure rather than supplemental figure 7.3. (If fewer main figures are needed, Figure 1 could be a supplement.)

We now include the CCA analysis as main Figure 8.

5. Ln 831 "Decoding of hand velocity from neural activity on control and PN stimulation trials": I appreciate the authors now asking whether kinematic decoding can improve by including DCN with M1 activity. The new data are said to show that this does not improve decoding over cortex alone. It would be helpful to follow the logic if the authors explained/accounted for the large difference in R-squared between the first analyses using the DCN or M1 alone with values at 0.55 vs those in the pooled case where the R2 is 0.78. It is difficult to appreciate the negative result with this baseline shift being so large.

The goodness-of-fit of the decoders tends to increase with the number of recorded neurons (for this analysis, multiunits). The sessions with simultaneous recordings in DCN and M1, which were chronologically the last and were spike-sorted with Kilosort 2, had high yields in M1 multiunits, so the baseline performance of the cortical decoder was relatively higher than most of the other sessions. We have added a sentence explaining this in the Methods. However, the difference in baseline decoder performance across sessions does not affect any central conclusions in the manuscript, and the effect of adding DCN activity to M1 on the decoding performance is limited across all 5 simultaneous recording sessions, even those with lower baseline performance (~0.65).

6. Discussion Paragraph line 448 – it should be mentioned that endpoints were hypermetric or hypometric with interposed activation and inactivation in Becker and Person 2019 and hypermetric with interposed ablation (Low.… Chen et al., 2018), not just that the velocities changed. Additionally, work in mice corroborated earlier studies in cats which showed endpoint problems with muscimol inactivation of the interposed nucleus (Cooper, Martin, Ghez 2000) which should be cited.

We have now added these references, and discuss the previous results.

7. Lines 450-452: The wording seems off: Rather than "taken together with these previous studies, our findings suggest" I would strongly recommend stating: "Consistent with these previous studies our findings support the idea that the cortico-cerebellar loop…" This recommendation is premised on the connotation of the original text that fine-tuning was not a conclusion previously drawn about cerebellar contribution to movement; to the contrary, it is a major and consistent theme, repeatedly brought up. The present data support those notions and are consistent with them.

We have re-worded this sentence along the lines suggested by the reviewer.

Reviewer #3 (Recommendations for the authors):The authors have made major revisions to the manuscript and have been largely responsive to reviewer comments. They now more clearly motivate the novelty of the present study in the context of the existing literature and have revised the Results to better align their concluding sentences with the actual results that are shown. The addition of histograms to Figures 5D-h are helpful in ore directly describing the effects of stimulation.To address reviewer concerns, the authors have also completed new analyses the latency of neural responses (Purkinje cells, DCN neurons, and motor cortical neurons) to optogenetic PN. I had assumed that quantification of latencies of the effects that are observes in the different nodes of the cortico-cerebellar loops would strengthen the authors' conclusion that they are actually studying the direct loop in Figure 1 which would then make the study's conclusions more compelling.As they detail in the results, the authors observe a logical increase in the latency of the activation along this pathway, for the small % of neurons (5%) with the shorted response times: 10ms for Purkinje cells, 13ms for DCN neurons, and 21 ms for cortical cells. They then acknowledge in their comments to the reviewers that the values – even for these fastest responding neurons – are longer than expected for pulse train stimulation. However, this point is not explained in the text.Additionally (and more importantly) the authors do not address the clear question arising from their new analysis, namely why do the vast majority of cells have such long response times to activation? For example, the new quantification of latencies added to the revision shows that the response latencies of most Purkinje cells are >63ms. Do the authors believe that such responses are mediated via a direct projections from the PN? If yes, more explanation is needed. If not, what are the alterative possible explanation(s)?. Either way, this point should be addressed in the revised Discussion.

As described in the Results, 50% of responsive Purkinje cells were recruited within 19 ms; the value of 63 ms refers to the motor cortical neurons.

We designed our latency analysis to detect the onset of the earliest responses to PN stimulation, and the choice of bin size required a tradeoff between temporal precision (requiring narrow time bins) and statistical power (requiring larger bins). We believe our choice (10ms) is a reasonable compromise, but of course some neurons might have been identified as non-responsive in our analysis which would be classified as responsive if the bin size were larger. Moreover, the fact that the stimulation was sinusoidal rather than a pulse train likely contributed to increasing the response latency. Hence the values of the earliest Purkinje cell responses we detect with our analysis likely underestimate the fastest responses, and we believe they are consistent with being mediated through feedforward excitation from the PN (via granule cells). The neurons in each region with long onset latencies (e.g., greater than 30ms) are instead presumably recruited through recurrent loops in the network, which we do not identify here.

We now state in the Results: “These values are likely underestimates of the fastest responses, as they were obtained using sinusoidal, rather than pulse train stimulation. The late-responding neurons are likely not recruited through feedforward excitation, but through unidentified, longer-latency feedback pathways.” In the Discussion, we state: “Neurons with long onset latencies (e.g., greater than 30 ms) are likely recruited through recurrent loops in the network, but our experiments do not enable us to identify these loops.”

Finally, as noted in my original review, the present these results are similar to those previously reported in primate DCN cooling experiments characterizing changes in hand movement in in a voluntary tracking task (e.g., Brooks et al., 1973; Conrad and Brooks 1974). The impact of the manuscript would be improved by returning to these studies in the discussion, and explaining why/how the application of the current methods provides a conceptual advance.

We now state in the Discussion: “Furthermore, by demonstrating that perturbation of a single source of mossy fiber input (which overlaps with inputs from sensory and other sources) impairs reaching, our work goes beyond previous studies documenting impairment from direct manipulation of the cerebellum itself (Brooks et al. 1973; Conrad et al. 1975; Meyer-Lohmann et al. 1975; Conrad and Brooks 1974; Becker and Person 2019).”

We emphasize that our results are not implied by these previous studies. It could have been the case, for example, that perturbation of pontocerebellar inputs alone would not have a large impact on reaching, and that simultaneous perturbation of pontocerebellar and cuneocerebellar inputs would be required to impair movement.